# HOLMEX: HUMAN-GUIDED SPURIOUS CORRELATION DETECTION AND BLACK-BOX MODEL FIXING

## ABSTRACT

We propose Holmex, a method for human-guided spurious correlation detection and black-box model fixing. Holmex provides a way for humans to be easily involved in the deep model debugging process, which includes 1) detecting conceptual spurious correlation in training data and 2) fixing biased black-box models by white-box models. In the first step, we leverage pre-trained vision-language model to construct separable vectors for some high-level and meaningful concepts, and we further propose a novel algorithm based on concept vectors that is more stable than previous methods. In the second step, unlike previous works, we do not constrain the original biased model to be interpretable and editable. Instead, Holmex is compatible with arbitrary black-box models. To this end, we propose transfer editing, a novel technique that can transfer the revision in interpretable models to the black-box models to correct their spurious correlations. Extensive experiments on multiple real-world datasets demonstrate the effectiveness of Holmex in detecting and fixing spurious correlations. The source code and datasets can be found in `https://anonymous.4open.science/r/Holmex-15DF`.

## 1 INTRODUCTION

Developing effective deep models for real-world problems is non-trivial, especially when the models make mistakes due to reliance on spurious correlations (Abid et al., 2022; Koh et al., 2020; Yuksekgonul et al., 2022). For example, using human face images from the web to classify gender might lead to a model with the stereotype that a person with long or special hair is more likely to be a woman. The developer needs to first find out the reason of hairstyle, and then fix it. In the common procedure, detecting the reason causing such mistake requires careful investigations on the model and training data, and deep understandings of the task, since the deep model is usually a black box. Thus, this *detecting process is costly and difficult* even for developers with rich machine learning expertise.

Moreover, after finding out the reason, *fixing these mistakes is also challenging* when not enough complementary data are available, and human knowledge becomes the ground to fix the mistake. This is the typical situation in the healthcare domain. Consider the scenario where chest X-rays are used to build a model to predict pneumonia. The model might lean on some spurious features caused by the data collection process, such as hospital-specific features (Zech et al., 2018). Further collecting data from another hospital is obstructed due to the high sensitivity of health data, so leveraging medical knowledge from doctors is necessary here. However, the gap between knowledge from doctors and the desired behaviors of the model is quite large. Filling the gap requires the developer to first fully understand medical knowledge and also tune the model very carefully. Overall, involving humans in the loop of deep model debugging and fixing is both knowledge-intensive and time-consuming.

In this paper, we introduce *Holmex* to tackle the above two difficulties of detecting and fixing. Holmex is a method for **H**uman-guided spurious c**O**rrelation detection and b**L**ack-box **M**od**E**l fi**X**ing. The two steps of detecting and fixing pose two requirements for our method, i.e., interpretability and editability. Specifically, our method should first detect and present correlations in the dataset to humans in an understandable and friendly way, and then provide an intuitive and easy-to-use method to fix the biased model. Note that Holmex aims at enabling humans to easily conduct spurious correlation detection and model fixing. Thus, humans determine what is a spurious correlation and where fixes are needed, while Holmex provides a convenient means of detection and fixing for humans.

For the first requirement about interpretability, we focus on concept-based methods, as they can provide human-friendly high-level conceptual interpretation. Finding the vector which represents a concept (e.g. "gender", "cat") in the embedding space is key to these methods. Early works (Kim et al., 2018; Abid et al., 2022) use some auxiliary datasets to build concept vectors, which is inflexible. Recent work (Yuksekgonul et al., 2022) leverages CLIP (Radford et al., 2021) to obtain concept vectors by encoding concept texts using CLIP's text encoder, and then simply use weights of the linear probe layer as correlations between concepts and labels. We show that raw text embeddings of concepts are entangled, and the above interpretations via weights of the linear layer are unstable. To address these issues, we propose a new concept vector by subtracting a vector of a background word. Also, we design a novel detecting algorithm to stably reveal correlations between concepts and labels.

About model fixing, different from previous works (Koh et al., 2020; Yuksekgonul et al., 2022; Bontempelli et al., 2023), Holmex does not constrain the original model to be interpretable, and it can be an arbitrary black-box model which is suitable for the problem. This setting is more practical and flexible since a large number of eep learning models are not specially designed to be interpretable. To this end, we propose the Transfer Editing technique that can transfer the revision in white-box models (i.e., interpretable models) to black-box models. Since those white-box models can be concept-level models, we can conduct concept-level fixing for black-box models. The contributions of this paper can be summarized as follows.

- For detecting, we improve the entangled raw text embeddings of concepts by subtracting a vector of the background word. Besides, we propose a novel detecting algorithm to stably reveal correlations between concepts and labels.

- For fixing, we propose the transfer editing technique that can transfer the revision made by humans in white-box models to black-box models, enabling black-box model fixing.

- Extensive experiments on multiple datasets with different biases (i.e., co-occurrence bias, picture style bias, and class attribute bias) are conducted to show the effectiveness of Holmex.

## 2 RELATED WORK

**Interpretable Deep Models with Concepts** Testing with Concept Activation Vectors (Kim et al., 2018) first proposed to interpret neural network's representations with high-level human-friendly concepts, such as "cat", and "brightness". Some subsequent research improves concept-based methods by aligning neurons with concept (Ribeiro et al., 2016; Koh et al., 2020; Chen et al., 2020), using large language models to generate concepts (Yang et al., 2023; Menon & Vondrick, 2022), incorporating vision-language model for better interpreting (Oikarinen et al., 2022; Oikarinen & Weng, 2022). These works focus on building interpretable deep models or explaining black-box models, while Holmex aims at both detecting and fixing for model development.

**Bias/Error Detection** Detecting model bias or error is critical for building robust models. Saliency map(Itti et al., 1998; Petsiuk et al., 2018; Wang et al., 2020; Zhang et al., 2021) is a kind of widely used visualization tool for explaining model behaviors and can be used for bias or error detection. This kind of method generally reflects the degree of importance of a pixel. Therefore, they cannot give high-level interpretations, like concepts. Conceptual Counterfactual Explanation (CCE) (Abid et al., 2022) combines concepts with counterfactual explanation methods (Wachter et al., 2017; Laugel et al., 2019; Mothilal et al., 2020), which use counterfactual examples for model interpretation. CCE interprets by correcting a mistake instance with concept level adjusting. CCE and many other detection works (Koh & Liang, 2017) are instance-level detection methods and also require instances in testing data. However, *our method leverages human knowledge, which does not need any testing data and can provide a class-level interpretation.*

**Model Fixing** Model fixing focuses on removing model errors. Some of them are designed to fix interpretable models. Specifically, Concept Bottleneck Model (CBM) (Koh et al., 2020) has a bottleneck layer where neurons are aligned with concepts. It can be edited by changing weights between concept bottleneck layer and the final prediction. Besides, prototypical part network (ProtoPNet) (Chen et al., 2019) focuses on finding prototypical parts and also supports debugging (Bontempelli et al., 2023). Those editing methods *can only be used in their specially designed white-box models, but our method does not have constraints on the model type.*

Domain adaptation (Farahani et al., 2021) can also be viewed as another branch of work for model fixing. These methods include invariant risk minimization (Arjovsky et al., 2019), domain adversarial neural network (Ajakan et al., 2014), gradient matching (Shi et al., 2021), and so on. Besides, instance reweighting (Amini et al., 2019) is a method that fixes bias by putting low importance on the samples that contain bias. A more recent work DISC (Wu et al., 2023) is a concept-level debugging method. However, these methods learn the bias by mining invariant features among different domains and they cannot help when we only have one domain. *In contrast, our method incorporates human knowledge and is for the scenario when there is only one domain.*

## 3 BACKGROUND

**Concept Bottleneck Model (CBM)** CBM (Koh et al., 2020) is a kind of interpretable deep model where one internal layer's neurons align with human-friendly concepts. CBM maps an input data $\boldsymbol{x}$ to a concept hidden layer $\boldsymbol{z}^c = g(\boldsymbol{x}) \in \mathbb{R}^m$ and bases predictions on these concept neurons, $y = f(\boldsymbol{z}^c)$. CBM needs the ground truth concept annotations for training. Suppose training data is $\{(\boldsymbol{x}_i, y_i, \boldsymbol{y}_i^c) | i \in [n]\}$ where $\boldsymbol{y}^c \in \mathbb{R}^m$ are labels for concepts. Then, its loss function can be denoted as $\mathcal{L} = \mathcal{L}_p(f(\boldsymbol{z}_i), y_i) + \lambda \mathcal{L}_a(\boldsymbol{z}_i, \boldsymbol{y}_i^c)$, where two terms are for label predicting and concept hidden layer constructing, respectively. To eliminate the need for concept annotations, a recent work called Post-hoc CBM (PCBM) (Yuksekgonul et al., 2022) uses a pre-trained vision-language model (illustrated below) to provide concept vectors.

**Contrastive Language-Image Pre-training (CLIP)** Pre-trained CLIP model builds the connection between languages and images, enabling zero-shot transfer to downstream tasks. CLIP has a text and a vision encoder, and they are optimized via contrastive learning. CLIP predicts the class label by calculating similarities between the embedding of label texts and images. Suppose there are $k$ candidate classes like {'cat', 'dog',..., 'airplane'}. CLIP maps those label words to some $d$ dimensional embedding vectors, which are $\{\boldsymbol{t}_{\text{cat}}, \boldsymbol{t}_{\text{dog}}, \dots, \boldsymbol{t}_{\text{airplane}}\}$, using its text encoder. For a given image $x$, CLIP maps it to an embedding vector $\boldsymbol{z} \in \mathbb{R}^d$ using the image encoder, and then makes a prediction as follows.

$$p(y = \text{cat} \mid \boldsymbol{z}) = \frac{\exp(\boldsymbol{t}_{\text{cat}}^\top \boldsymbol{z})}{\exp(\boldsymbol{t}_{\text{cat}}^\top \boldsymbol{z}) + \exp(\boldsymbol{t}_{\text{dog}}^\top \boldsymbol{z}) + \cdots + \exp(\boldsymbol{t}_{\text{airplane}}^\top \boldsymbol{z})} \tag{1}$$

Thanks to the large amount of training data and large model that CLIP used, the performance of CLIP in a zero-shot setting is still quite well, which makes the possible to extract concepts from CLIP.

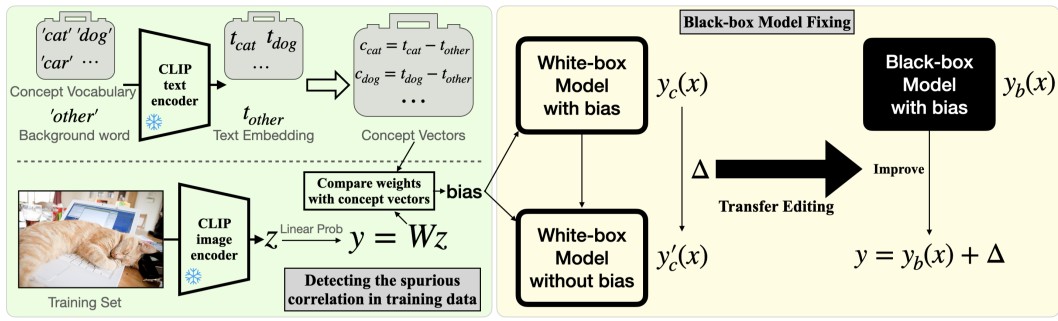

Figure 1: Illustration of detecting the spurious correlation in training data and black-box model fixing.

## 4 DETECTING SPURIOUS CORRELATIONS IN TRAINING DATA

In this section, we present how Holmex can help detect the spurious correlation in training data, which is shown in the left part in Figure 1. Holmex provides interpretable information on correlations between concepts and each class label. Then, humans can check if some unreasonable concepts have too strong correlations in the classification model, and thus detect spurious correlations.

Generally, the detecting process can be divided into two steps, i.e., constructing concept vectors and revealing the correlations between concepts and labels. They are illustrated in the following two subsections, respectively. For better understanding, we will discuss our method under a detailed task shown in Figure 2.

## 4.1 Constructing concept vectors

The basic idea is to leverage CLIP to construct concept vectors. Suppose there is a concept vocabulary that contains $m$ concept words, and let $\{t_1, t_2, \ldots, t_m\}$ be the CLIP text embeddings of them. $x$ denotes an image and $z \in \mathbb{R}^d$ is the CLIP image embedding of $x$. The previous work (Yuksekgonul et al., 2022) uses raw text embeddings as concept vectors and uses $t_i^\top z$ to indicate the activation of concept $i$ shown in the image. Following Yuksekgonul et al. (2022), we also do not apply any prompt like "a photo of {}". Since Eq. (1) can be a good classifier among those concepts, it seems natural to use $t_i$ as the concept vector. However, we shall argue its issues as follows.

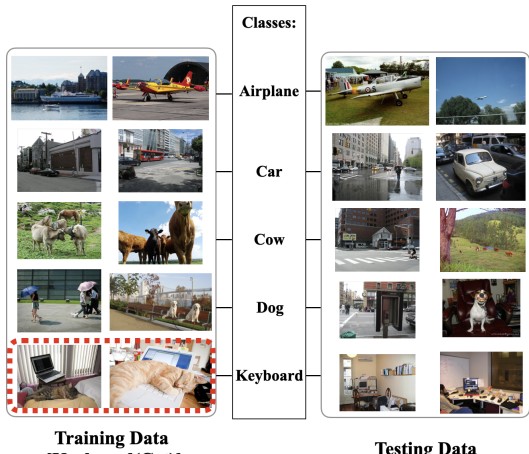

### 4.1.1 Disadvantages of raw text embedding

We find that the raw text embedding $t_i$ cannot disentangle those concepts well. As shown in Figure 3(a), "cat" and "airplane" seem to be two irrelevant words, while the cosine-similarity between their text embeddings is about 0.74 which is quite large. Furthermore, we conduct a model

Figure 2: Samples in training and testing data. The keyboard images in the training set always contain a cat, while there is almost no cat appearing in testing keyboard images. So the concept "cat" is the spurious concept for the class "keyboard".

editing experiment based on raw concept vectors. We train a linear layer after the concept activation layer composed by $t_i^\top z, i \in [m]$, and then edit the model via setting weights of the spurious concept in the linear layer to 0. The experiment details can be found in Appendix A.1 and we report the average results in Table 1. The performance degeneration after model editing further demonstrates that the raw text embeddings are not good concept vectors.

### 4.1.2 Subtracting a background concept vector

The above issues are alleviated by our proposed concept vectors, which are derived by incorporating embeddings from CLIP into the method in TCAV (Kim et al., 2018). Concept vectors in TCAV are obtained by training a binary linear classifier to distinguish between activations produced by a concept's example images and other images in the bottleneck layers of a network, and then using the weight of the linear classifier as the concept vector of the corresponding concept. Here, we leverage CLIP to build such binary classifier. For instance, if we want to use CLIP to decide whether an image has the concept "cat", we should use at least two words, "cat" and a background word. Here, we discuss more about the background word.

**Background word** The principle of choosing the background word is that the background word should not be relevant to any concept words. Thus, we should use a meaningless word that does not imply any specific entity. We make an ablation study in Appendix B, and we find that using words like "other", "a", "that", "else" have similar performance on the downstream tasks. In the following context, we just choose the "other" as the background word.

After we choose a background word, CLIP decides the score of the "cat" concept as follows.

$$p(y = \text{cat} \mid z) = \frac{\exp(t_{\text{cat}}^\top z)}{\exp(t_{\text{cat}}^\top z) + \exp(t_{\text{other}}^\top z)} = \sigma((t_{\text{cat}} - t_{\text{other}})^\top z),$$

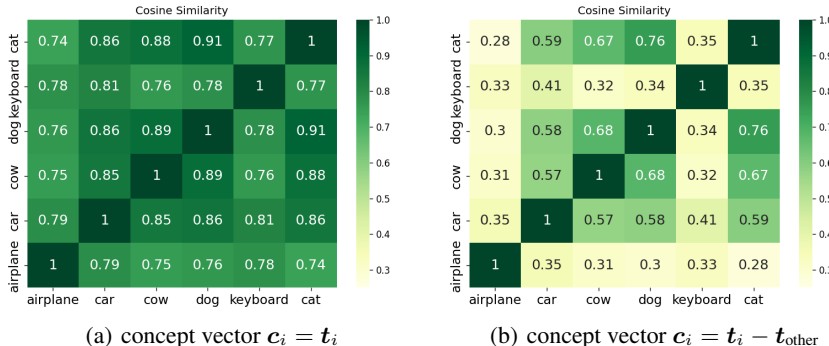

(a) concept vector $c_i = t_i$        (b) concept vector $c_i = t_i - t_{\text{other}}$

Figure 3: The cosine-similarity between concept vectors. On the left is the result when we just use the raw text embedding as the concept vectors, e.g. $c_{\text{cat}} = t_{\text{cat}}$. On the right is the result when we use our purposed concept vectors, e.g. $c_{\text{cat}} = t_{\text{cat}} - t_{\text{other}}$. The CLIP model we used is ViT-B/32. Similar results when using different CLIP models are shown in Figure 6.

where $\sigma$ is the sigmoid function. We notice that $\sigma((t_{\text{cat}} - t_{\text{other}})^\top z)$ is the classification result for the bottleneck layer $z$, and $t_{\text{cat}} - t_{\text{other}}$ is the classifier weights. Thus, we suggest using $t_{\text{cat}} - t_{\text{other}}$ as the concept vector. To illustrate the advantages of our proposed concept vectors, we first show in Figure 3 that our concept vectors have better disentanglement for those concepts that are pretty different from each other. Second, we conduct the same model editing experiment as the last subsection with our proposed concept vectors. Results in Table 1 show that by using the subtracted concept vectors, we can improve model performance by editing.

Table 1: The testing accuracy before and after editing when we use different concept vectors.

| Concept vector | $c_i = t_i$ | | | (Ours) $c_i = t_i - t_{\text{other}}$ | | |
|---|---|---|---|---|---|---|
| Editing | before | after | increase | before | after | increase |
| Average accuracy | 84.17% | 71.64% | -12.53% | 83.62% | 84.67% | 1.05% |

## 4.2 REVEALING CORRELATIONS BETWEEN CONCEPTS AND LABELS

After obtaining the concept vectors, we use them to reveal correlations between concepts and labels in data to humans. A straightforward way adopted by Yuksekgonul et al. (2022) is to use the weight of the linear classifier built in the concept activation layer. Specifically, the concept activation layer $z^c \in \mathbb{R}^m$ consists of the activations of all concepts $z_i^c = c_i^\top z$, $i \in [m]$, where $z$ is the embedding from CLIP image encoder of the input $x$ and $c_i$ is the $i^{\text{th}}$ concept vector. Then the weight matrix $W \in \mathbb{R}^{k \times m}$ in the final softmax function $\mathbf{softmax}(Wz^c)$ for multiclass classification is considered as the correlation matrix between class labels and concepts. However, we show that such a method may not output stable interpretations based on the following facts.

**Fact 1 (Multicollinearity of $z^c$)** *When $m$ is larger than the embedding dimension $d$, some concept vectors must be linearly dependent, and $z^c$ must be multicollinear (Gujarati & Porter, 2003). Such multicollinearity makes the learned weight matrix $W$ non-identifiable, which means there exist at least two different $W$ leading to the same loss when learning $\mathbf{softmax}(Wz^c)$.*

In practice, the embedding dimension $d$ is usually pre-defined[1] while we often need to test a large number of candidate concepts due to potential unknown bias. Thus, it is very common that $m > d$.

**Fact 2 (Invariance to Constant Shift)** *For any constant vector $\beta \in \mathbb{R}^m$, $W$ and $W' = W + [\beta, \beta, \dots, \beta]^\top$ have the same training loss when learning $\mathbf{softmax}(Wz^c)$.*

---

[1] In CLIP, the embedding dimensions of ViT-B/32 and ViT-L/14 are 512 and 768, respectively.

Both Fact 1 and Fact 2 imply that using the learned weight matrix $W$ to reveal correlations between class labels and concepts may not give us stable interpretations, as there often exists multiple equivalent weight matrices when learning $\textbf{softmax}(Wz^c)$. Note that if we sort concepts for a specific class label, the importance rank of concepts may vary for two equivalent but different $W$.

We give two examples in Table 6 and 7 in the Appendix to illustrate the two kinds of instability risks caused by Fact 1 and Fact 2, respectively.

### 4.2.1 STABLE DETECTION OF CORRELATIONS

To stabilize the detected correlation between class labels and concepts, we propose a novel method shown in Algorithm 1 to address the above issues. In general, we do not use the concept activation layer $z^c$ and the weight of the linear layer upon it to reflect correlations. Instead, we directly use the image embedding $z$ as the feature vector to train a multiclass logistic model $\textbf{softmax}(Wz)$, and treat $w_l$ (the $l^{\text{th}}$ row of $W$) as the representation of the $l^{\text{th}}$ class, similar to existing methods such as TCAV (Kim et al., 2018). Then we use $c_i^\top w_l - \frac{1}{k}\sum_{l'=1}^{k} c_i^\top w_{l'}$ as the correlation between concept $i$ and class label $l$. The advantages are as follows.

- Using $z$ instead of $z^c$ can avoid the multi-collinearity issue in Fact 1.

- The calculated correlation $c_i^\top w_l - \frac{1}{k}\sum_{l'=1}^{k} c_i^\top w_{l'}$ stays the same even when we add a constant shift $\beta \in \mathbb{R}^d$ to all the $w_l$'s. Therefore, the issue caused by Fact 2 is also resolved.

---

**Algorithm 1** Detecting spurious concepts

**Require:** $m$ concept vectors $c_i = t_i - t_{\text{other}}, i \in [m]$
**Require:** Training data $\mathcal{S} = \{(x_j, y_j) \mid j \in [n]\}$
**Require:** Embeddings $z_j = encoder(x_j) \in \mathbb{R}^d$
    **Train linear model:**
1: Obtain $W \in \mathbb{R}^{k \times d}$ by training the multiclass logistic regression model $\textbf{softmax}(Wz)$ on $\mathcal{S}$
    **Record interpretations:**
2:   $recorder \leftarrow \{\}$      ▷ a list for recording
3: **for** $i \in [m]$ **do**     ▷ iterate through all concepts
4:     $\zeta_i \leftarrow \arg\max_l c_i^\top w_l$   ▷ concept $c_i$ correlates most with class $\zeta_i$
5:     $\alpha_i \leftarrow c_i^\top w_{\zeta_i} - \frac{1}{k}\sum_{l=1}^{k} c_i^\top w_l$   ▷ correlation strength
6:     Add tuple $(c_i, \zeta_i, \alpha_i)$ to $recorder$
7: **end for**
8: Let humans check if there are spurious correlations in $recorder$

---

As human checking may still be costly, for each concept $i$, we only care about the class label $\zeta_i$ that $i$ benefits the most (line 2∼7). After figuring out every concept's most correlated class and correlation strength, we let human experts check if there are spurious correlations and detect spurious concepts. For example, if we find the concept "cat" can benefit class "keyboard" with extremely large strength, then people can detect this spurious concept. An example of the recorder produced by Algorithm 1 for the task in Fig. 2 can be found in Fig. 9 and Fig. 10 in Appendix.

## 5 BLACK-BOX MODEL FIXING

With spurious correlations detected in the last section, we now present the transfer editing technique, enabling fixing arbitrary black-box models via transferring the revision in white-box models to black-box models. This is illustrated in the right part in Figure 1.

### 5.1 TRANSFER EDITING

As illustrated in Figure 1, transfer editing contains two steps. The first one is building two white-box models to capture the influence of removing the spurious correlation. The second step is transferring the difference between the two while-box models to the biased black-box model to fix it.

**Step 1: Build unbiased and biased white-box models.** The first step is shown in Figure 4. Instead of achieving good quantitative model performance, our goal in this step is capturing the influence of removing the bias (spurious correlation). The spurious concept is detected by humans with the assistance of our detecting process, so the editing direction is guided by humans. The two white-box models we adopt are built by frozen CLIP backbone and linear function based on concept activation layers $z^c$. For the white-box model without bias, we only use the class labels as concepts. For the white-box model with bias, we also include the spurious concept in the concept set. To further boost our method, we give some practical suggestions on the choices of the two white-box models in Appendix C.3.

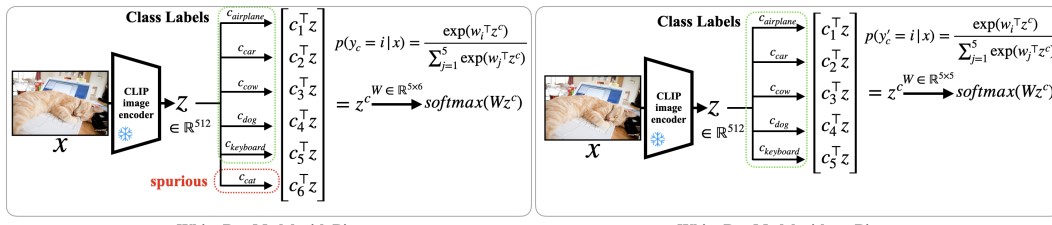

Figure 4: Illustration of two white-box models used to capture the influence of removing the spurious correlation in the task shown in Figure 2.

**Step 2: Transfer the difference.** In the second step, we move the difference in logits of the two white-box models to the target black-box model. Specifically, let $y_c(\boldsymbol{x})$ and $y'_c(\boldsymbol{x})$ be the output before the final softmax function, i.e., logits, of the white-box model with bias and without bias, respectively, and $y_b(\boldsymbol{x})$ be the logits of any black-box model which is also trained on the same training data. Since those three models are all for the same task, their logits are in the same space and can be linearly combined. The goal is to transfer the difference between the two white-box models to the black-box model. Therefore, we simply use $\Delta = y'_c(\boldsymbol{x}) - y_c(\boldsymbol{x})$ to capture effect caused by the spurious correlation. Then, adding $\Delta$ to $y_b(\boldsymbol{x})$, which results in $y_b(\boldsymbol{x}) + \Delta$, can remove the effect of bias in the black-box model.

**Why logit space instead of probability space?** There are two design principles of modifying $y_b$ by $y_c$ and $y'_c$: **(1)** the output should be a valid probability distribution $p = [p_1, \ldots, p_k]$ where $p_i \geq 0$, for $i \in [k]$, and $\sum p_i = 1$; and **(2)** when $y'_c = y_c$, the final output should be the same as the black-box model's original output. We have tried several other plans such as operating in the probability space. However, only the method described in Step 2 of Transfer Editing satisfies those two design principles. More explanations are put in Appendix C.1.

**The scale of logits.** Suppose we have two models $f(x)$ and $g(x) = cf(x)$. Then $f(x)$ and $g(x)$ have the same argmax prediction, while $f(x) + \Delta$ and $g(x) + \Delta$ may have different argmax predictions. Although $f(x)$ and $g(x)$ have the same argmax prediction, their probability distributions over class labels are different. Note that the distribution over classes indicates the confidence that a sample belongs to a specific class. One important intuition is that a normal model should output a proper probability distribution over class labels. As a result, if the model does not output probabilities too different from the proper ones, applying $y_b + \Delta$ should keep logits on the right scale. We discuss this more in Appendix C.2.

## 5.2 Combining transfer editing and ensemble

Ensemble is a useful machine learning technique that also combines different models similar to transfer editing. Here, we first explain their difference, and further provide a way to combine them. The success of ensemble learning relies on the good performance and diversity of the base models, while transfer editing works due to bias removal with the help of human prior knowledge. Moreover, ensemble learning and transfer editing can be combined as follows. We first ensemble the black box $y_b(\boldsymbol{x})$ and the white box with bias $y_c(\boldsymbol{x})$ by model averaging. Since both $y_b(\boldsymbol{x})$ and $y_c(\boldsymbol{x})$ contain the spurious correlation in training data, the ensemble of those two models should also remain biased. Thus, we can further apply transfer editing $\Delta$ to get a better result. In Appendix C.4, we show that ensemble before and after transfer editing are equivalent.

## 6 Experiments

We test three different types of bias for comprehensive experiments listed below. More details about the datasets can be found in Appendix D.

**Co-occurrence bias.** Following Yuksekgonul et al. (2022), we use the Meta-Shift dataset (Liang & Zou, 2021) and construct eight classification tasks. In each task, there are five classes and one of

them has spurious co-occurred entities in training data, while testing data contains no bias. Figure 2 shows an example that in each image of "keyboard", there is always a cat nearby in training data.

**Picture style bias.** We use the Office-Home (Venkateswara et al., 2017) dataset to build two tasks containing the bias of picture style. For the first task, in the training set, all the bike's pictures are in CLIP art style while the pictures of other classes are real-world images. In the testing set, pictures of bike return to normal and others are changed to CLIP art style. The second task is also a similar task.

**Class attribute bias.** We use the Celeb-A (Liu et al., 2015) dataset which contains images of human faces to build the gender classification task with class attribute bias. We build a task that in the training set, all males have black hair while females have hair in other colors. In the testing set, the attribute of hair color is reversed. This task implies a typical stereotype in real-world situations, where a person with special hair is more likely to be a woman.

## 6.1 DETECTING SPURIOUS CONCEPTS

In this section, we compare our proposed detection method with other baseline approaches across eight co-occurrence bias tasks. For all tasks and methods, we utilized the same concept vocabulary collected from the Chinese college entrance examination English syllabus. We limit our focus to verbs and nouns, resulting in a total of 1,953 words. The word list can be found in our source code repository. We now introduce the compared methods and the metrics we used.

**Methods in Comparison.** The first baseline is **PCBM** (Yuksekgonul et al., 2022) which is the latest and SOTA concept-level CLIP-based detection method. In PCBM, a linear model $y = W z^c$ is learned using training data, and $z^c \in \mathbb{R}^{1953}$. PCBM uses the weight between the class (e.g. "keyboard") and every concept as the correlation strength. The second baseline is **PCBM-improve**, where after $W$ in PCBM is obtained, the weight between the concept and class minus the average weight of that concept is used as the correlation strength. PCBM-improve keeps a consistent interpretation under $W$ and $W + [\beta, \ldots, \beta]^\top$ for any $\beta \in \mathbb{R}^m$. The third baseline is **Ours (raw embedding)** which is the same as ours except the concept vector is raw texting embedding. For all methods, we sort concepts related to a class (e.g. "keyboard") by correlation strength and present the sorted concept list to humans.

**Evaluation and Results.** We sort the concepts related to a class (e.g. "keyboard") by the correlation strength and evaluate the results using the rank of the ground truth bias concept (e.g. "cat"). The average results across all eight tasks are listed in Table 2. Our detecting method is the best among all baselines. PCBM-based methods do not perform well since there are 1,953 concepts in this task, much larger than the dimension of the embedding (i.e., 512). PCBM-improve is better than PCBM as it addresses the second issue in section 4.2. Using raw embedding as the concept vector also causes performance degeneration, as we expected.

Table 2: The average result among all eight tasks on detecting the spurious concept. The rank is **smaller**, and the method is **better**. The detailed scores for each class are in Table 12 in the Appendix.

| Method | PCBM(SOTA) | PCBM-improve | Ours (raw embedding) | Ours |
|---|---|---|---|---|
| average rank $\pm$ std | 1128.7$\pm$342.7 | 317.9$\pm$326.0 | 74.4$\pm$255.2 | **16.7$\pm$5.4** |

## 6.2 FIXING BLACK-BOX MODELS

In this section, we test our proposed transfer editing technique on various black-model fixing tasks. The experiments are conducted as follows: Initially, black-box models are trained on the training sets outlined at the beginning of section 6. These models are influenced by spurious correlations, as the training sets consist of such. Consequently, we train two white-box models and perform transfer editing on the black-box model to fix it. We then test the accuracy on test sets that do not contain any spurious correlations. The model details are described as follows.

**Model Architectures.** We describe model architectures by their backbone and the following classifier. For all tasks, we used three backbones which are CLIP ViT-L/14, CLIP ViT-B/32 and ImageNet (Deng et al., 2009) pre-trained Resnet50 (He et al., 2016). All the backbones are frozen and we only build classifiers on the output embeddings of the backbone. The white-box models we used are described in section 5.1, while the classifier for black-box models is an MLP.

**Results and Analysis.** From the results in Table 3 and Table 4, we have the following findings.

**(1) Transfer editing helps.** Results show that the performance of $y_b + \Delta$ is better than the original black box $y_b$. Table 3 reports results when using different backbones, indicating that transfer editing remains beneficial regardless of the backbone used. Similar findings can be obtained from results on tasks with style bias and attribute bias (Table 25 and 26 in Appendix). Besides, we try a fine-tuned Resnet50 as our black-box model, and in this case, the performance of $y_b + \Delta$ is still better than $y_b$, which means transfer editing is also helpful when black-box models are fine-tuned models (Table 24 in Appendix). Moreover, we check the accuracy of the biased class and find that our method dramatically increases its accuracy (Appendix D.3). To sum up, those results indicate that transfer editing can help improve black-box models on various tasks and scenarios.

**(2) Transfer editing is different from ensemble.** Results show that transfer editing + ensemble performs best among all methods. We also examine the results if we ensemble by $\lambda y_b + (1 - \lambda)y_c$ in Appendix D.5. Although results show $\lambda = 1/2$ is of course not always the optimal for different datasets, we find that no matter which $\lambda$ is used, the performance of $\lambda y_b + (1-\lambda)y_c + \Delta$ is consistently better than $\lambda y_b + (1 - \lambda)y_c$. These results indicate that transfer editing is different from ensemble and can be used together to obtain a better performance. We do not discuss how to find the optimal $\lambda$, since it is beyond the scope of this work. Besides, we also compare our method with zero-shot CLIP, and it outperforms CLIP by 2% (Appendix D.4). Although zero-shot CLIP does not contain bias, it cannot utilize the training data either.

Table 3: The testing accuracy for co-occurrence bias tasks when we use different black-box models.

| | $y_c$ | $y'_c$ | $y_b$ ViT-L/14 | transfer edit +ensemble | transfer edit | $y_b$ ViT-B/32 | transfer edit +ensemble | transfer edit | $y_b$ Resnet50 | transfer edit +ensemble | transfer edit |
|---|---|---|---|---|---|---|---|---|---|---|---|
| Average | 0.8362 | 0.8681 | 0.8369 | **0.8795** | 0.8599 | 0.8271 | **0.8762** | 0.8383 | 0.8175 | **0.8770** | 0.8161 |
| Task1 | 0.8580 | 0.9026 | 0.9049 | **0.9271** | 0.9249 | 0.8943 | **0.9237** | 0.8994 | 0.8746 | **0.9261** | 0.8880 |
| Task2 | 0.7787 | 0.8402 | 0.7803 | **0.8435** | 0.8190 | 0.7693 | **0.8389** | 0.7922 | 0.7783 | **0.8492** | 0.7850 |
| Task3 | 0.8117 | 0.8416 | 0.7872 | **0.8415** | 0.8079 | 0.7837 | **0.8434** | 0.8019 | 0.7781 | **0.8379** | 0.7741 |
| Task4 | 0.8231 | 0.8289 | 0.8030 | **0.8349** | 0.8059 | 0.7957 | **0.8364** | 0.7943 | 0.7545 | **0.8253** | 0.7583 |
| Task5 | 0.7911 | 0.8702 | 0.8009 | **0.8914** | 0.8662 | 0.7888 | **0.8816** | 0.8157 | 0.7909 | **0.8860** | 0.7660 |
| Task6 | 0.8903 | 0.8982 | 0.8998 | **0.9138** | 0.9023 | 0.8734 | **0.9028** | 0.8771 | 0.8645 | **0.9130** | 0.8662 |
| Task7 | 0.8824 | 0.8926 | 0.8773 | **0.9008** | 0.8873 | 0.8560 | **0.9001** | 0.8649 | 0.8479 | **0.8973** | 0.8519 |
| Task8 | 0.8544 | 0.8703 | 0.8421 | **0.8831** | 0.8654 | 0.8554 | **0.8823** | 0.8606 | 0.8510 | **0.8812** | 0.8394 |

Table 4: Testing accuracy of Picture Style Bias and Class Attribute Bias. The backbone of white-box and black-box are ViT-B and ViT-L.

| | $y_c$ | $y'_c$ | $y_b$ | transfer edit +ensemble | transfer edit | $\frac{y_b+y_c}{2}$ | $\frac{y_b+y'_c}{2}$ | $\frac{y_b+y_c+y'_c}{3}$ |
|---|---|---|---|---|---|---|---|---|
| Picture Style Bias | 0.9254 | 0.9458 | 0.9290 | **0.9752** | 0.9518 | 0.9549 | 0.9681 | 0.9551 |
| Class Attribute Bias | 0.9732 | 0.9819 | 0.9852 | **0.9883** | 0.9856 | 0.9860 | 0.9876 | 0.9857 |

**More analytical experiments.** **(1) When black-box models do not suffer from bias.** We conduct the same experiment where we simply replace the biased training data with another training data that does not contain spurious correlations. In this case, Transfer Editing fails to improve the black-box models, so the conclusion is no bias results in no help. This experiment further supports that Transfer Editing improves the model accuracy in the way of removing bias (see Appendix D.6). **(2) When using less training data for white-box models.** Sometimes, accessing the entire training data may be difficult or time-consuming. Thus, we consider the setting that we can only use 5% training data to train white-box models. Under such circumstances, the performances of all white-box models are worse than black-box models and our method can still work (see Appendix D.7). **(3) When a part of the data contains spurious correlations.** In this experiment, 90% of the keyboard images have "cat" as the spurious concept and the other 10% images are unbiased. We find that transfer editing still works and beats another baseline under this setting. (see Appendix D.8)

## 7 CONCLUSION

In this paper, we present Holmex which can be helpful for humans in detecting the bias in the training set and fixing any black-box models. Holmex constructs less entangled concept vectors compared to previous works, and further employs a novel spurious correlation detecting algorithm that is stable even when a large number of concepts are utilized. Moreover, we propose the transfer editing technique that can transfer the revision in interpretable models to black-box models, enabling fixing arbitrary black-box models. Experiments on real-world datasets well demonstrate that Holmex can be helpful for deep model debugging and fixing in practical scenarios.

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

# A    ISSUES OF PCBM

In this section, we illustrate issues of PCBM (Yuksekgonul et al., 2022) via some experimental results.

## A.1    ADVANTAGE OF OUR PROPOSED CONCEPT VECTORS

We first show that our proposed concept vectors can perform better than those of PCBM if we edit CBM by erasing the weights of the spurious concept.

We treat each class label as a concept and we use in total six concepts, including all class labels as well as the spurious concept. Then we map each image to image embedding by CLIP image encoder and map the image embedding to six concept neurons by dot-product with those concept vectors. To further make a classification prediction, we use a linear weight that maps those six concepts to five classes. Figure 5 shows the weight after training and editing under the task in Figure 2.

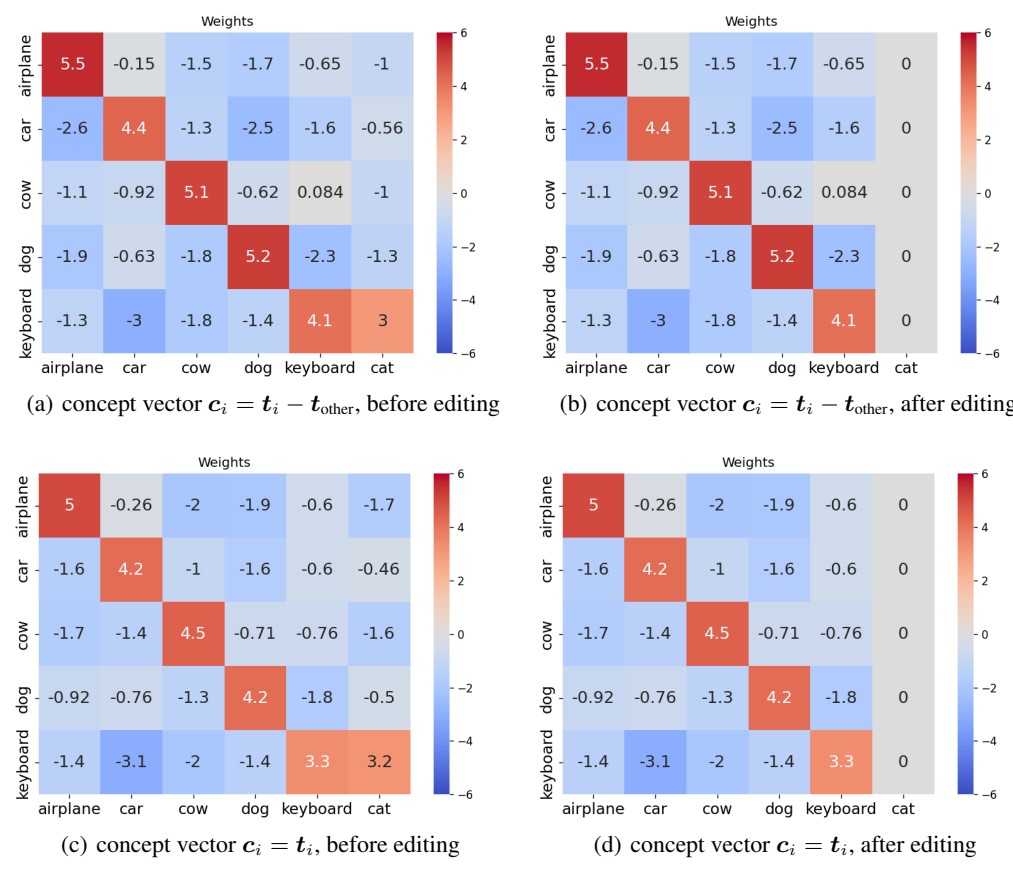

(a) concept vector $c_i = t_i - t_{\text{other}}$, before editing  (b) concept vector $c_i = t_i - t_{\text{other}}$, after editing

(c) concept vector $c_i = t_i$, before editing  (d) concept vector $c_i = t_i$, after editing

Figure 5: Weight matrix before and after editing. We edit the model by setting the weight of the spurious concept as 0. The y-axis corresponds to classes, and the x-axis corresponds to concepts. Here we use 6 concepts which are the text of labels and spurious concept cat.

We conduct experiments on eight tasks (the tasks' information can be found in Table 10) to compare the model performance before and after we remove the impact of the spurious concept. Table 5 shows the results, where the model performance always degrades sharply after editing when using concept vectors of PCBM. However, if we use our proposed concept vectors $c_i = t_i - t_{\text{other}}$, removing the impact of the spurious concept only compromise the testing accuracy a little or even can improve the testing accuracy. Such results demonstrate the advantage of our proposed concept vectors compared to the concept vectors of PCBM.

Table 5: The testing accuracy before and after editing when we use different types of concept vectors. All experiments are averaged on five random seeds. The CLIP model we use is ViT-L/14.

| Concept Vector | $c_i = t_i - t_{\text{other}}$ | | | $c_i = t_i$ | | |
|---|---|---|---|---|---|---|
| editing | before | after | increase | before | after | increase |
| Average | 83.62% | 84.67% | 1.05% | 84.17% | 71.64% | -12.53% |
| Task1 | 85.80% | 90.42% | 4.62% | 85.78% | 71.48% | -14.30% |
| Task2 | 77.87% | 81.93% | 4.06% | 78.78% | 69.89% | -8.89% |
| Task3 | 81.17% | 79.38% | -1.79% | 81.58% | 72.13% | -9.45% |
| Task4 | 82.31% | 79.57% | -2.74% | 82.21% | 70.67% | -11.54% |
| Task5 | 79.11% | 83.97% | 4.86% | 82.15% | 71.04% | -11.11% |
| Task6 | 89.03% | 87.94% | -1.09% | 89.02% | 74.67% | -14.35% |
| Task7 | 88.24% | 88.80% | 0.56% | 88.39% | 74.23% | -14.16% |
| Task8 | 85.44% | 85.35% | -0.09% | 85.44% | 69.01% | -16.43% |

A potential reason for the superiority of our proposed concept vectors is that our concept vectors can better distinguish between different concepts, as reported in Figure 3. To further verify this, we adopt another CLIP model to retest the disentanglement of concept vectors and report the results in Figure 6, which are very similar to Figure 3.

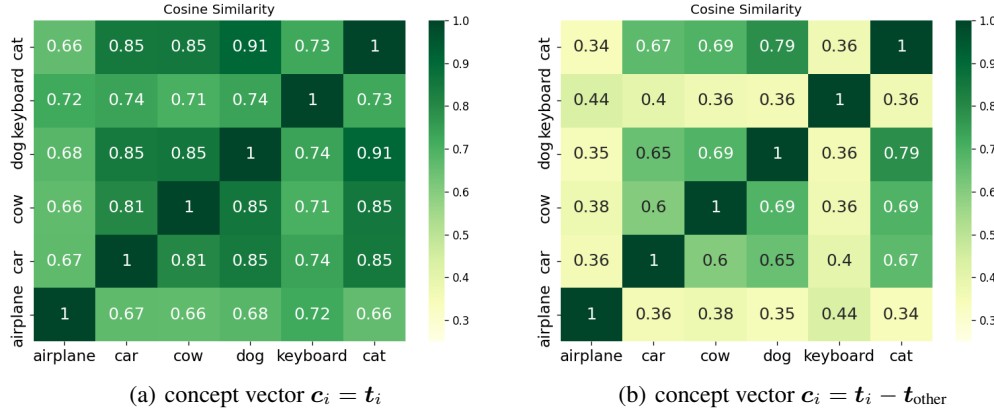

(a) concept vector $c_i = t_i$        (b) concept vector $c_i = t_i - t_{\text{other}}$

Figure 6: The cosine-similarity between concept vectors. On the left is the result when we just use the raw text embedding as the concept vectors, e.g. $c_{\text{cat}} = t_{\text{cat}}$. On the right is the result when we use our purposed concept setting, e.g. $c_{\text{cat}} = t_{\text{cat}} - t_{\text{other}}$. The CLIP model we use is ViT-L/14.

## A.2 UNSTABLE PCBM DETECTION

We give two concrete examples showing that the detection by PCBM can be unstable. Table 6 presents two equivalent (w.r.t. training loss) but different weight matrices due to Fact 1. Table 7 is an example of two equivalent (w.r.t. training loss) but different weight matrices due to Fact 2. Note that in both Table 6 and Table 7, the two "equivalent" weight matrices lead to drastically different interpretations of the importance of concepts to class labels.

Table 6: An example of two weight matrices having the same training loss due to Fact 1. This example illustrates the risk of linear dependency under PCBM detecting setting. We use the simplest linear dependency setting where two concept vectors are identical. Here we assume the concepts "feline" and "cat" are identical and as a result, the upper weight matrix and the lower weight matrix have the same training loss. However, these two "equivalent" weight matrices have very different interpretations. The upper weight matrix under PCBM setting implies that for the class "keyboard", the concept "cat" is very important while "feline" is not important at all. The lower weight matrix indicates the opposite conclusion. Since linear dependency is inevitable when we want to detect many more concepts, the detecting method in PCBM is not suitable for large concept sets.

| Class \ Concepts | Tab key | Fur | engine | feline | cat | ... | ... |
|---|---|---|---|---|---|---|---|
| Airplane | 1 | -3 | 6 | ... | ... | ... | ... |
| Car | 2 | -4 | 8 | ... | ... | ... | ... |
| Cow | -1 | 2 | -3 | ... | ... | ... | ... |
| Dog | 0 | 9 | -1 | ... | ... | ... | ... |
| keyboard | 8 | 3 | 1 | 1 | 99 | ... | ... |
| Airplane | 1 | -3 | 6 | ... | ... | ... | ... |
| Car | 2 | -4 | 8 | ... | ... | ... | ... |
| Cow | -1 | 2 | -3 | ... | ... | ... | ... |
| Dog | 0 | 9 | -1 | ... | ... | ... | ... |
| keyboard | 8 | 3 | 1 | 99 | 1 | ... | ... |

Table 7: An example of two weight matrices having the same training loss due to Fact 2. The lower weight matrix is obtained by adding a constant vector $\beta = (-100, 0, 0, \ldots, \ldots)$ to each row of the upper weight matrix. According to the softmax function, these two matrices have the same training loss. If we directly use the weight to interpret the importance of a concept to a class label, the two "equivalent" weight matrices give drastically different interpretations of how important the concept "Tab key" is to the class "keyboard", as the weight in the upper matrix is 8 and the weight in the lower one is -92. This example again shows that directly using the weight for detection can be unstable.

| Class \ Concepts | Tab key | Fur | engine | ... | ... |
|---|---|---|---|---|---|
| Airplane | 1 | -3 | 6 | ... | ... |
| Car | 2 | -4 | 8 | ... | ... |
| Cow | -1 | 2 | -3 | ... | ... |
| Dog | 0 | 9 | -1 | ... | ... |
| keyboard | 8 | 3 | 1 | ... | ... |
| Airplane | -99 | -3 | 6 | ... | ... |
| Car | -98 | -4 | 8 | ... | ... |
| Cow | -101 | 2 | -3 | ... | ... |
| Dog | -100 | 9 | -1 | ... | ... |
| keyboard | -92 | 3 | 1 | ... | ... |

## B  THE BACKGROUND WORD

The principle of choosing the background word is that the background word should be not relevant to the concept word. Thus we should choose those words that are meaningless. Here we consider four words which are 'other', 'a', 'that', and 'else'. We conduct the detection experiments on those background words. The results in Table 8 shows that there are no dramatic difference between those four words and they are all better than the case if we do not use any background words.

Table 8: The performance of different background words. The average rank of spurious concept. The number is smaller, the background word is better.

| background word | 'other' | 'a' | 'that' | 'else' | no background word |
|---|---|---|---|---|---|
| rank | 16.7 | 21.9 | 25.2 | 16.5 | 74.4 |
| std | 5.4 | 40.7 | 46.4 | 4.6 | 255.2 |

We notice that in the segmentation area, (Li et al., 2022) used the background word "other" to perform semantic segmentation. For example, suppose the three words ['dog', 'tree', 'other'] are used, then the pixels of dogs are assigned to the first class, pixels of trees are assigned to the second class and all other pixels are assigned to the last class. Thus we choose the background word 'other' instead of the word 'else' to keep the same setting as another research region.

Lastly, we also consider using the average embedding as the background word embedding. We conducted an experiment to test such a background word for detection, where the average embedding background achieves an average rank of 140 which is worse than "other" with an average rank of 16.7.

## C  BLACK-BOX MODEL FIXING

### C.1  HOW TO COMBINE THE WHITE-BOX MODELS AND BLACK-BOX MODELS

There are two principles when we choose the combination of models.

- First, it should always output a valid probability distribution $p = [p_1, p_2, p_3, \ldots, p_k]$ where $p_i \geq 0$, for $i \in [k]$, and $\sum_{i=1}^{k} p_i = 1$.
- Second, when $y'_c = y_c$, the final output should be the same as that of the black-box model.

Let $y$ be the output in logit space and $d = \text{softmax}(y)$ in the probability space. To give a final probability output, we have the following choices.

1. Let $y_o = y_b + y_{c'} - y_c$ and we output $d_o = \textbf{softmax}(y_o)$. This method is what we adopt in the paper and it satisfies both principles.

2. We first convert all model output to probability space and output $d = d_b + d_{c'} - d_c$. This method could not guarantee the first principle. Consider an example $d_b = (0.50, 0.49, 0.01), d_{c'} = (0.50, 0.50, 0.00), d_c = (0.50, 0.48, 0.02)$, and $d = d_b + d_{c'} - dc = (0.50, 0.51, -0.01)$.

3. Let $\Delta = y_{c'} - y_c$ and $d_\Delta = \textbf{softmax}(\Delta)$, we then output $d = d_b + d_\Delta$. This one also violates the first principle because $\sum p_i = 2$ in this method.

4. We output $d = \frac{d_b + d_\Delta}{2}$. This one violates the second principle. For example $d_b = (0.8, 0.1, 0.1), \Delta = (0, 0, 0)$, then

$$d = \frac{d_b + d_\Delta}{2} = \frac{(0.8, 0.1, 0.1) + (1/3, 1/3, 1/3))}{2} = (17/30, 13/60, 13/60) \neq d_b$$

In summary, only the first one satisfies both principles, thus we adopt it.

### C.2  THE SCALE OF LOGITS

Suppose we have two models $f(x)$ and $g(x) = cf(x)$, where $c > 1$. Then $f(x)$ and $g(x)$ have the same argmax prediction, while $f(x) + \Delta$ and $g(x) + \Delta$ may have different argmax predictions.

Although $f(x)$ and $g(x)$ have the same argmax prediction, it does not mean that they are equivalent to each other. This is because the probability distributions output by $f(x)$ and $g(x)$ are different. Please note that the probability distribution indicates the confidence that a sample belongs to a specific class. Consider a weather forecasting task where we want to output the probability of rainy and sunny. Suppose $f(x) = (0.1, 0)$, which means the prediction distribution is $(0.525, 0.475)$. This means the

model actually cannot predict whether it is going to rain or not with a very high confidence level as the two probabilities are too close to each other. If $g(x) = 100 f(x) = (10, 0)$, the prediction probability distribution becomes $(0.999955, 0.000045)$, which means that the model thinks it is going to rain with an extremely high confidence. Therefore, it is reasonable that $f(x)$ and $g(x) = c f(x)$ could make different predictions after removing the effect of bias as the original confidence levels of the two models are pretty different from each other.

We discuss more about the scale of logit. One important intuition is that a normal model should output a proper probability distribution over class labels. For example, if there are 80% rainy samples and 20% sunny samples for the same given input $x$ in the training set, then a normal model will output a probability distribution similar to $(0.8, 0.2)$. Since the mapping between probabilities and logits is bijective (if we ignore the constant shift), we can derive that the logits are roughly $(1.375 + constant, 0 + constant)$ based on the probabilities $(0.8, 0.2)$. Thus, a normal model's logits can be roughly decided by the proper probabilities. As a result, if the model does not output probabilities too different from the proper ones, applying $y_b + \Delta$ should keep logits on the right scale.

## C.3  OUR SUGGESTIONS ON THE TWO WHITE-BOX MODELS

We give some suggestions on the two white-box models for deriving the impact of removing spurious correlations. Note that the two white-box models in the model fixing stage can be different from the white-box used in the detection phase. In fact, to better derive the impact of removing spurious correlations, we suggest using different white-box models in this stage.

Recall Figure 4. Once the white-box model without bias is fixed, the other one with bias is also fixed. Therefore, the key to choosing two proper white-box models is to figure out a proper white-box without bias. Note that for a white-box model, the most important feature is the concept set, so our major job is to choose a good set of concepts. We illustrate two intuitive requirements on the concept set as follows.

1. The good concepts (all except for the spurious one) should benefit the classification task.

2. The number of good concepts should not be too large as we want to avoid multicollinearity.

Based on the above requirements, a very natural choice is to directly use all class labels as good concepts, since class labels themselves are the most informative features for classification and the number of classes is usually not too big. Therefore, in our experiments, we train a new CBM to obtain the white-box model with bias, where the concept set contains all class labels and the spurious concept identified in the detection phase. Note that one can use other concept sets as long as the two requirements above are satisfied.

Once the white-box model with bias is trained, we have two natural ways to obtain the white-box model without bias. The first choice is to edit the white-box model with bias by zeroing out the weight of the spurious concept in the layer before the final softmax function. The other way is to train a new white-box model by setting the concept set as the set of only good concepts. We conduct experiments to check which way is better. We call the white-box model obtained via the first way **CBM-edit** and the one obtained via the second way **CBM-clean**. Besides the two models, we also train another CBM on a training set where we replace the images of the corrupted class with another set of images to make the training set and the testing set follow the same distribution. As such CBM totally has no spurious correlation issue, its accuracy can be regarded as a rough upper bound of the effectiveness of any CBM without bias.

The experimental results on the Meta-Shift tasks described in Table 10 are shown in Table 9. We find that CBM-clean is generally more effective than CBM-edit. Moreover, the performance of CBM-clean is even very close to that of the CBM trained on the unbiased data, which implies that the CBM-clean is probably effective enough. Therefore, in our experiments, we adopt CBM-clean as the white-box model without bias.

Table 9: Performance of training a new clean CBM and editing on biased CBM. The backbone we used is CLIP-ViT/14, and all the experiments are run for five random seeds. The concepts are class labels and the spurious concept. CBM-edit is obtained by setting the weight of the spurious concept in CBM to zero. CBM-clean is a new model that only uses class labels as concepts, and it is trained on the original training set from scratch.

|  | CBM trained on unbiased data | CBM | CBM-edit | CBM-clean |
|---|---|---|---|---|
| Average | 86.99% | 83.62% | 84.67% | 86.81% |
| Task1 | 90.78% | 85.80% | 90.42% | 90.26% |
| Task2 | 84.65% | 77.87% | 81.93% | 84.02% |
| Task3 | 83.89% | 81.17% | 79.38% | 84.16% |
| Task4 | 83.79% | 82.31% | 79.57% | 82.89% |
| Task5 | 87.15% | 79.11% | 83.97% | 87.02% |
| Task6 | 89.47% | 89.03% | 87.94% | 89.82% |
| Task7 | 89.45% | 88.24% | 88.80% | 89.26% |
| Task8 | 86.73% | 85.44% | 85.35% | 87.03% |

### C.4 Ensemble + Transfer Editing

To combine transfer editing with ensemble learning, we can first ensemble the black box $y_b(\boldsymbol{x})$ and the white box with bias $y_c(\boldsymbol{x})$. Since both $y_b(\boldsymbol{x})$ and $y_c(\boldsymbol{x})$ contain the bias in training data, the assembled model of those two models should also have the bias. Thus, we can further apply the transfer editing $\Delta$ to get a better model.

$$y_b(\boldsymbol{x}) \xrightarrow{\text{Ensemble}} \frac{y_b(\boldsymbol{x}) + y_c(\boldsymbol{x})}{2} \xrightarrow{\text{Transfer Editing}} \frac{y_b(\boldsymbol{x}) + y_c(\boldsymbol{x})}{2} + \Delta = \frac{y_b(\boldsymbol{x})}{2} + y'_c(\boldsymbol{x}) - \frac{y_c(\boldsymbol{x})}{2}$$

Another way to combine transfer editing with ensemble learning is to first apply transfer editing $\Delta$ to $y_b(\boldsymbol{x})$, and then we ensemble the unbiased model $y_b(\boldsymbol{x}) + \Delta$ with the white box without bias $y'_c(\boldsymbol{x})$. We show below that this leads to the same model as when we do ensemble learning first.

$$y_b(\boldsymbol{x}) \xrightarrow{\text{Transfer Editing}} y_b(\boldsymbol{x}) + \Delta \xrightarrow{\text{Ensemble}} \frac{y_b(\boldsymbol{x}) + \Delta + y'_c(\boldsymbol{x})}{2} = \frac{y_b(\boldsymbol{x})}{2} + y'_c(\boldsymbol{x}) - \frac{y_c(\boldsymbol{x})}{2}$$

Similarly, those two ways of combination are also the same when we ensemble two models by weighted sum under the weight $(\lambda, 1 - \lambda)$.

## D More Details on Experiments

### D.1 Detailed information of datasets

The first task of our study used the Meta-Shift dataset (Liang & Zou, 2021) to present the bias of co-occurrence of entities. In this co-occurrence bias task, we constructed eight classification tasks. In each task, there are five classes and one class of data contains a spurious correlation. Every image in the training data of that class contains the same co-occurring entity, but there is no co-occurrence in the testing set. Distributions of training data of other classes are the same as the testing data. The detailed classes and co-occurring entities are listed in Table 10.

Table 10: Information of tasks in the Meta-Shift dataset. The class labels are shown in brackets and the co-occurring concepts are shown in parentheses.

| Task Number | Class Label | #training sets | #testing sets |
|---|---|---|---|
| Task1 | ['airplane', 'car', 'cow', 'dog', 'keyboard(cat)'] | 665 | 4000 |
| Task2 | ['bed', 'car(snow)', 'cat', 'computer', 'motorcycle'] | 980 | 3000 |
| Task3 | ['airplane', 'bed', 'bus', 'car(America)', 'dog'] | 380 | 5000 |
| Task4 | ['airplane', 'bed', 'cat', 'computer', 'couch(glasses)'] | 680 | 3000 |
| Task5 | ['bed(cat)', 'bird', 'computer', 'dog', 'motorcycle'] | 950 | 3000 |
| Task6 | ['bus', 'cat', 'computer', 'pillow(clock)', 'snowboard'] | 505 | 2500 |
| Task7 | ['bus', 'cat', 'computer', 'snowboard', 'television(fireplace)'] | 520 | 2750 |
| Task8 | ['airplane', 'bed(dog)', 'bus', 'car', 'cow'] | 685 | 4000 |

The second task of our study used the Office-Home (Venkateswara et al., 2017) dataset to present the bias of picture style. We used the Office-Home dataset to build two tasks. Office-Home contains many classes with different picture styles. For the first task, we selected five classes which are "Alarm Clock", "Bike", "Candles", "Fan" and "Trash Can". In the training set, all the bike pictures are in CLIP art style while the pictures of other classes are real-world images. In the testing set, the styles of all classes are reversed. Some instances of pictures are shown in Figure 7. The second task is similar, wherein the training set, images of the class "Backpack" are real-world ones, while images of class "Bottle", "Chair", "Computer" and "Hammer" are in CLIP art style. In the testing set, the styles are inverse.

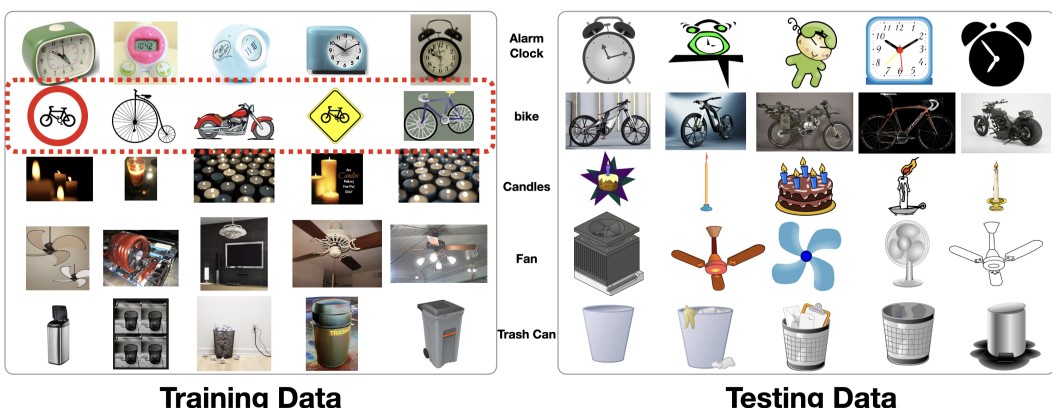

Figure 7: Some instances in the task for picture style bias.

The last task of our study focuses on the bias of attributes of classes. For this kind of bias, we leveraged the Celeb-A dataset (Liu et al., 2015). The goal is gender classification using human face images. In the training set, all males have black hair while females have other colors of hair. In the testing set, the attributes of hair color are reversed. There are 40,000 images in the training and testing set. Some example images of this task are shown in Figure 8. We also show statistics on the relationship between black hair and gender in the original Celeb-A dataset in Table 11, which implies that the correlation between black hair and gender does exist.

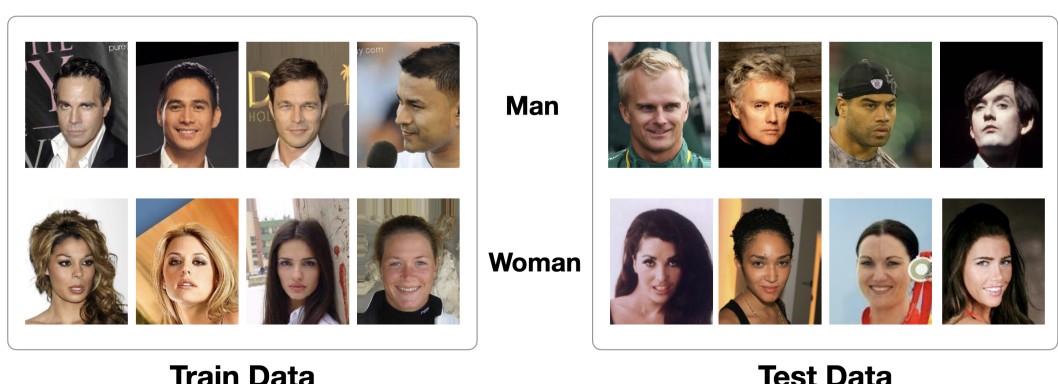

Figure 8: Some instances in the task for class attribute bias.

Table 11: Statistical Check on the original Celeb-A dataset. The proportion of black hair in males is larger than in females.

| | Hair Color | | |
| Gender | Black | Other | Total by Gender |
|---|---|---|---|
| Male | 25, 156 | 59, 278 | 84, 434 |
| Female | 23, 316 | 94, 849 | 118, 165 |
| Total by Hair Color | 48, 472 | 154, 127 | 202, 599 |

### D.2    EXTRA FIGURES AND TABLES IN DETECTION EXPERIMENTS

Here we show some internal results generated by our detection algorithm. Figure 9 shows the different concepts generated by the algorithm and the class in which they are most helpful, as well as the correlation strength of that class. In Figure 10, we filter out all the concepts in Figure 9 where the second column is the "keyboard", and we can see that the "cat" appears in the first row of Figure 9, so we can easily detect this spurious concept.

```
 1 : ['airplane', 'airplane', 0.2741368763148785]      41 : ['bird', 'airplane', 0.11713417079299689]
 2 : ['plane', 'airplane', 0.2564011227339506]         42 : ['skyscraper', 'airplane', 0.1148956086486578]
 3 : ['cow', 'cow', 0.24514553919434548]               43 : ['settlement', 'cow', 0.11475476771593093]
 4 : ['aircraft', 'airplane', 0.24083200953900813]     44 : ['chat', 'keyboard', 0.1143839119002223]
 5 : ['cattle', 'cow', 0.23658954873681068]            45 : ['ceiling', 'airplane', 0.11371464189141989]
 6 : ['airport', 'airplane', 0.21570854708552362]      46 : ['parrot', 'airplane', 0.11305700577795505]
 7 : ['flight', 'airplane', 0.19821176677942276]       47 : ['assistant', 'keyboard', 0.11171569488942623]
 8 : ['airline', 'airplane', 0.19663245175033808]      48 : ['pilot', 'airplane', 0.1111761206710025]
 9 : ['dog', 'dog', 0.17754030898213385]               49 : ['parking', 'car', 0.11083869785070419]
10 : ['cat', 'keyboard', 0.16260749772191047]          50 : ['air', 'airplane', 0.11081477720290422]
11 : ['computer', 'keyboard', 0.16177478656172753]     51 : ['zebra', 'cow', 0.11025844737887383]
12 : ['desk', 'keyboard', 0.15807506758719683]         52 : ['typewriter', 'keyboard', 0.10847749412059784]
13 : ['hamburger', 'cow', 0.1577414296567744]          53 : ['mud', 'cow', 0.10810531377792358]
14 : ['agriculture', 'cow', 0.15770503897219895]       54 : ['citizen', 'airplane', 0.1077938076108694]
15 : ['airmail', 'airplane', 0.15660716444253922]      55 : ['mutton', 'cow', 0.10778596214950084]
16 : ['monitor', 'keyboard', 0.15198333449661733]      56 : ['arrival', 'airplane', 0.10646355673670768]
17 : ['farm', 'cow', 0.14881090652197598]              57 : ['eagle', 'airplane', 0.1063492932356894]
18 : ['farmer', 'cow', 0.1480724962428212]             58 : ['toothpaste', 'airplane', 0.10622899048030376]
19 : ['fly', 'airplane', 0.1429894270375371]           59 : ['grass', 'cow', 0.10588411763310432]
20 : ['beef', 'cow', 0.14118442395702005]              60 : ['seagull', 'airplane', 0.1050791334360838]
21 : ['village', 'cow', 0.1389554798603058]            61 : ['departure', 'airplane', 0.10466452278196811]
22 : ['ox', 'cow', 0.1360697922296822]                 62 : ['peasant', 'cow', 0.10405174493789673]
23 : ['hay', 'cow', 0.13383483309298755]               63 : ['milk', 'cow', 0.10389852151274681]
24 : ['countryside', 'cow', 0.1331622201949358]        64 : ['prejudice', 'cow', 0.10377251338213682]
25 : ['car', 'car', 0.13247295264154674]               65 : ['habit', 'cow', 0.10372884348034858]
26 : ['sky', 'airplane', 0.1315979329869151]           66 : ['service', 'airplane', 0.10369235295802355]
27 : ['kindergarten', 'cow', 0.13127943985164164]      67 : ['orbit', 'airplane', 0.1036747332662344]
28 : ['India', 'cow', 0.13085923120379447]             68 : ['office', 'keyboard', 0.10364451184868813]
29 : ['novelist', 'keyboard', 0.13071177005767823]     69 : ['astronaut', 'airplane', 0.10317778438329697]
30 : ['butcher', 'cow', 0.1279236269183457]            70 : ['receptionist', 'keyboard', 0.10315879434347153]
31 : ['typist', 'keyboard', 0.12749477960169314]       71 : ['Greece', 'airplane', 0.1026765551418066]
32 : ['jet', 'airplane', 0.12669914923608303]          72 : ['conservation', 'cow', 0.1026191383600235]
33 : ['tractor', 'cow', 0.1259258132427931]            73 : ['wool', 'cow', 0.10208905637264251]
34 : ['bark', 'dog', 0.1257800734601915]               74 : ['hawk', 'airplane', 0.10139948194846511]
35 : ['keyboard', 'keyboard', 0.12500782534480095]     75 : ['piano', 'cow', 0.10113855078816414]
36 : ['wing', 'airplane', 0.12484468519687653]         76 : ['fence', 'cow', 0.10107250977307558]
37 : ['helicopter', 'airplane', 0.1235603891313076]    77 : ['mouse', 'keyboard', 0.09978157319128514]
38 : ['mosquito', 'airplane', 0.11922845542430878]     78 : ['cab', 'car', 0.09853741712868214]
39 : ['secretary', 'keyboard', 0.11825886592268944]    79 : ['hunger', 'cow', 0.09842476751655341]
40 : ['sheep', 'cow', 0.11745587978512048]             80 : ['country', 'cow', 0.0981886301189661]
```

Figure 9: An example record obtained by our detection algorithm for the example task in Figure 2. Each row contains a tuple. For example, "['wing', 'airplane', '0.1248']" means that the term "wing" is most helpful for the class "airplane" with a correlation strength of 0.1248.

```
10 :  ['cat', 'keyboard', 0.16260749772191047]          219 :  ['degree', 'keyboard', 0.07520209103822709]
11 :  ['computer', 'keyboard', 0.16177478656172753]     220 :  ['wire', 'keyboard', 0.07514765597879887]
12 :  ['desk', 'keyboard', 0.15807506758719683]         227 :  ['function', 'keyboard', 0.07452280037105083]
16 :  ['monitor', 'keyboard', 0.15198333449661733]      230 :  ['businesswoman', 'keyboard', 0.07424316070973873]
29 :  ['novelist', 'keyboard', 0.13071177005767823]     235 :  ['table', 'keyboard', 0.07402810007333756]
31 :  ['typist', 'keyboard', 0.12749477960169314]       238 :  ['pencil', 'keyboard', 0.07394250631332397]
35 :  ['keyboard', 'keyboard', 0.12500782534480095]     248 :  ['homework', 'keyboard', 0.0732308093458414]
39 :  ['secretary', 'keyboard', 0.11825886592268944]    251 :  ['fur', 'keyboard', 0.07307719178497792]
44 :  ['chat', 'keyboard', 0.1143839119002223]          264 :  ['button', 'keyboard', 0.07175233326852322]
47 :  ['assistant', 'keyboard', 0.11171569488942623]    267 :  ['compromise', 'keyboard', 0.07156877145171166]
52 :  ['typewriter', 'keyboard', 0.10847749412059784]   275 :  ['reporter', 'keyboard', 0.07053797105327249]
68 :  ['office', 'keyboard', 0.10364451184868813]       278 :  ['ash', 'keyboard', 0.07040706928819418]
70 :  ['receptionist', 'keyboard', 0.10315879434347153  284 :  ['virus', 'keyboard', 0.06978425607085229]
77 :  ['mouse', 'keyboard', 0.09978157319128514]        285 :  ['carbon', 'keyboard', 0.06978139877319336]
85 :  ['writing', 'keyboard', 0.09730694591999053]      302 :  ['biscuit', 'keyboard', 0.0691514689475298]
89 :  ['editor', 'keyboard', 0.09688697196543217]       305 :  ['collar', 'keyboard', 0.0689558274112641]
95 :  ['diploma', 'keyboard', 0.0950447466224432]       309 :  ['cup', 'keyboard', 0.0688264936208725]
97 :  ['Internet', 'keyboard', 0.09476110009716988]     315 :  ['apartment', 'keyboard', 0.06848681773990392]
98 :  ['colleague', 'keyboard', 0.0944951904937625]     322 :  ['pear', 'keyboard', 0.06794457621872425]
101 :  ['staff', 'keyboard', 0.09335615672171116]       328 :  ['deadline', 'keyboard', 0.06764245368540286]
103 :  ['vase', 'keyboard', 0.0927316516637802    1]      330 :  ['string', 'keyboard', 0.067480250261724]
111 :  ['librarian', 'keyboard', 0.08999386802315712]   331 :  ['applicant', 'keyboard', 0.06746201068162919]
113 :  ['dictation', 'keyboard', 0.08962374366819859]   336 :  ['drawer', 'keyboard', 0.06721668392419815]
115 :  ['videophone', 'keyboard', 0.08910688571631908]  337 :  ['eraser', 'keyboard', 0.06704179346561431]
117 :  ['translator', 'keyboard', 0.08822774868458509]  341 :  ['lap', 'keyboard', 0.06680112443864346]
119 :  ['tiger', 'keyboard', 0.08805085979402065]       344 :  ['companion', 'keyboard', 0.06668550465255976]
124 :  ['bedroom', 'keyboard', 0.08761225696653127]     349 :  ['author', 'keyboard', 0.06638449784368276]
127 :  ['journalist', 'keyboard', 0.08694722056388855]  352 :  ['certificate', 'keyboard', 0.06628505364060402]
132 :  ['operator', 'keyboard', 0.08554942328482866]    358 :  ['speaker', 'keyboard', 0.0658782966434957]
134 :  ['printer', 'keyboard', 0.08515811152756214]     373 :  ['pillow', 'keyboard', 0.06492962948977947]
140 :  ['revision', 'keyboard', 0.08399844244122505]    387 :  ['mineral', 'keyboard', 0.0642131470143795]
147 :  ['paperwork', 'keyboard', 0.0815247118473053]    388 :  ['coal', 'keyboard', 0.0641868045553565]
153 :  ['scholar', 'keyboard', 0.0807006236165761    9]   390 :  ['chair', 'keyboard', 0.06413454972207547]
158 :  ['page', 'keyboard', 0.08029622361063957]        392 :  ['assistance', 'keyboard', 0.06394921727478504]
173 :  ['loaf', 'keyboard', 0.07958234213292599]        401 :  ['equipment', 'keyboard', 0.0633807385340333]
179 :  ['bamboo', 'keyboard', 0.07848974578082561]      407 :  ['apple', 'keyboard', 0.06322363540530204]
191 :  ['opera', 'keyboard', 0.07711814697831869]       409 :  ['scarf', 'keyboard', 0.06318373195827007]
210 :  ['cushion', 'keyboard', 0.0760398855702229]      419 :  ['directory', 'keyboard', 0.06278369873762131]
213 :  ['technology', 'keyboard', 0.07559903860092163]  427 :  ['interpreter', 'keyboard', 0.0625925324857235]
218 :  ['clerk', 'keyboard', 0.07525377683341503]       428 :  ['notebook', 'keyboard', 0.06256471201777458]
```

Figure 10: We select all the tuples which is most helpful for "keyboard" in Figure 9.

Table 12: The detailed results on detecting experiments. The CLIP backbone we used is ViT/B-32.

| Method | PCBM | PCBM-improve | Ours (raw embedding) | Ours |
|---|---|---|---|---|
| average rank | 1128.7 | 317.9 | 74.4 | **16.7** |
| average std | 342.7 | 326.0 | 255.2 | **5.4** |
| Task1 rank | 1432.6 | 5.4 | 157.6 | 1.6 |
| Task1 std | 391.2 | 5.0 | 532.1 | 0.9 |
| Task2 rank | 1703.4 | 1.0 | 1.0 | 1.0 |
| Task2 std | 257.9 | 0.2 | 0.0 | 0.0 |
| Task3 rank | 146.5 | 188.0 | 1.2 | 1.2 |
| Task3 std | 238.4 | 494.1 | 0.6 | 0.5 |
| Task4 rank | 783.9 | 1492.5 | 182.5 | 110.9 |
| Task4 std | 470.9 | 762.3 | 365.3 | 37.7 |
| Task5 rank | 1600.1 | 1.2 | 1.0 | 1.0 |
| Task5 std | 323.6 | 0.4 | 0.0 | 0.0 |
| Task6 rank | 277.8 | 631.7 | 104.6 | 7.3 |
| Task6 std | 306.3 | 851.9 | 426.2 | 2.0 |
| Task7 rank | 1462.7 | 216.2 | 66.4 | 8.1 |
| Task7 std | 374.9 | 485.2 | 333.5 | 1.4 |
| Task8 rank | 1622.4 | 7.4 | 80.6 | 2.8 |
| Task8 std | 378.2 | 8.5 | 384.1 | 0.8 |

## D.3 WORST-GROUP ACCURACY

Table 13 shows the accuracy of the biased class(e.g. class "keyboard" in Figure 2). As you can see the transfer editing method dramatically increases the accuracy of biased class.

Table 13: The average testing accuracy of biased class.

| Worst class | Task1 | Task2 | Task3 | Task4 | Task5 | Task6 | Task7 | Task8 |
|---|---|---|---|---|---|---|---|---|
| $y_b$ | 0.8475 | 0.3300 | 0.4080 | 0.5653 | 0.3940 | 0.7628 | 0.6175 | 0.8040 |
| $y_b + \Delta$ | 0.9653 | 0.6737 | 0.5501 | 0.6090 | 0.8750 | 0.7824 | 0.7062 | 0.9418 |

## D.4 ZERO-SHOT CLIP

Table 14 shows the accuracy of zero-shot CLIP and the accuracy of our method(transfer edit + ensemble). Our method outperforms the zero-shot CLIP by 2%. Although zero-shot CLIP does not contain bias, it can not utilize the training data either.

Table 14: The average testing accuracy of Zero-shot CLIP and Ours.

| Tasks | Task1 | Task2 | Task3 | Task4 | Task5 | Task6 | Task7 | Task8 | Average |
|---|---|---|---|---|---|---|---|---|---|
| Zero-shot CLIP | 0.8935 | 0.8240 | 0.8272 | 0.8387 | 0.8720 | 0.8780 | 0.8851 | 0.8580 | 0.8596 |
| Ours | 0.9271 | 0.8435 | 0.8415 | 0.8349 | 0.8914 | 0.9138 | 0.9008 | 0.8831 | **0.8795** |

## D.5 EXPERIMENT WITH DIFFERENT LAMBDA

First, we emphasize that we do not claim that $(y_b + y_c)/2 + \Delta$ is the best combination and $(y_b + y_c)/2$ is the best ensembling way. This paper is not aiming to find out which $\lambda$ makes the best ensembling $(\lambda y_b + (1 - \lambda)y_c)$, since it is the scope of ensembling literature. What we want to convey in this paper is that no matter what $\lambda$ is used to make an ensembling $\lambda y_b + (1 - \lambda)y_c$, we can always improve it by adding $\Delta$. The reason is $y_b$ and $y_c$ both contain bias and $\Delta$ can remove this bias. Table 15 clearly shows that under different choices of $\lambda$, our transfer editing always helps.

Table 15: The results of task 1 for different $\lambda$.

| $\lambda$ | 0 | 0.1 | 0.2 | 0.3 | 0.4 | 0.5 | 0.6 | 0.7 | 0.8 | 0.9 | 1 |
|---|---|---|---|---|---|---|---|---|---|---|---|
| $\lambda y_b + (1 - \lambda)y_c$ | 85.73 | 86.79 | 87.74 | 88.53 | 89.24 | 89.77 | 90.04 | 90.25 | 90.49 | 90.50 | 90.49 |
| $\lambda y_b + (1 - \lambda)y_c + \Delta$ | 90.25 | 91.06 | 91.43 | 92.06 | 92.50 | 92.72 | 92.88 | 92.93 | 92.93 | 92.78 | 92.50 |
| improvement | 4.53 | 4.27 | 3.69 | 3.53 | 3.26 | 2.95 | 2.84 | 2.68 | 2.44 | 2.28 | 2.01 |

We also check the results of $\lambda y_b + (1 - \lambda)y_{c'}$ in Table 16. $\lambda y_b + (1 - \lambda)y_{c'}$ has two part, the second part $(1 - \lambda)y_{c'}$ does not contain bias while the first part $\lambda y_b$ still has bias. Thus $\lambda y_b + (1 - \lambda)y_{c'}$ can be always improved by $\lambda(y_b + \Delta) + (1 - \lambda)y_{c'}$. Again, we do not want to discuss how to find the best ensembling method. Our point is after we have a suitable ensembling result, applying transfer editing on it will always boost the performance.

Table 16: The results of $\lambda y_b + (1 - \lambda)y_{c'}$.

| $\lambda$ | 0 | 0.1 | 0.2 | 0.3 | 0.4 | 0.5 | 0.6 | 0.7 | 0.8 | 0.9 | 1 |
|---|---|---|---|---|---|---|---|---|---|---|---|
| $(\lambda y_b + (1 - \lambda)y_{c'})/2$ | 90.25 | 90.92 | 91.36 | 91.73 | 91.69 | 91.83 | 91.69 | 91.64 | 91.34 | 90.99 | 90.49 |
| $(\lambda(y_b + \Delta) + (1 - \lambda)y_{c'})/2$ | 90.25 | 91.06 | 91.43 | 92.06 | 92.50 | 92.72 | 92.88 | 92.93 | 92.93 | 92.78 | 92.50 |
| improvement | 0.00 | 0.14 | 0.07 | 0.34 | 0.81 | 0.89 | 1.19 | 1.29 | 1.60 | 1.79 | 2.01 |

## D.6 NO BIAS, NO HELP

We conducted another experiment to show that if the black boxes do not suffer from the bias, then Transfer Editing will not help. For co-occurrence bias tasks, we replace the images of the biased

class with normal images, so that training data have the same distribution as testing data. We further train the black-box models on the training data to obtain $y_b^*$ which do not suffer from bias. Then we try to use the white-box models trained on original training data to conduct transfer editing for $y_b^*$. Although white-box models can capture the bias, $y_b^*$ is not influenced by the bias. The results shown in Table 17 illustrate that transfer editing does not help if the black-box model is not influenced by the bias. Those results further confirm that transfer editing works by removing bias.

Table 17: The average testing accuracy when we use an unbiased training set to train black-box models. White-box models are still trained by the original training set.

|         | $y_c$  | $y_c'$ | $y_b^*$ | transfer edit |
|---------|--------|--------|---------|---------------|
| Average | 0.8362 | 0.8681 | 0.8776  | 0.8542        |
| Task1   | 0.8580 | 0.9026 | 0.9257  | 0.9208        |
| Task2   | 0.7787 | 0.8402 | 0.8507  | 0.7989        |
| Task3   | 0.8117 | 0.8416 | 0.8275  | 0.8099        |
| Task4   | 0.8231 | 0.8289 | 0.8427  | 0.8268        |
| Task5   | 0.7911 | 0.8702 | 0.8937  | 0.8137        |
| Task6   | 0.8903 | 0.8982 | 0.9099  | 0.9097        |
| Task7   | 0.8824 | 0.8926 | 0.9016  | 0.8939        |
| Task8   | 0.8544 | 0.8703 | 0.8689  | 0.8595        |

## D.7 LESS TRAINING DATA

In this section, we only use 5% of training data to train white-box models and use CLIP ViT-B/32 as the backbone of white-box models. Here we first present the result of the keyboard (cat) experiments(Table 10, task 1) in Table 18.

Table 18: The testing accuracy of task1 when the white-box models are training by 5% training data.

| keyboard(cat) | $y_c$ | $y_{c'}$ | $y_b$ | $y_b + \Delta$ | $(y_b + y_c)/2$ | $(y_b + y_c)/2 + \Delta$ |
|---------------|--------|----------|--------|----------------|-----------------|--------------------------|
| accuracy      | 0.8043 | 0.8781   | 0.9049 | 0.9208         | 0.9033          | 0.9261                   |

When the $y_c$ and $y_{c'}$ are weaker than the black-box model $y_b$, the transfer editing method can also improve the original $y_b$ or the simple ensembling $(y_b + y_c)/2$. Here we present the result of task 7 "television(fireplace)" in Table 19.

Table 19: The testing accuracy of task7 when the white-box models are training by 5% training data.

| television(fireplace) | $y_c$ | $y_{c'}$ | $y_b$ | $y_b + \Delta$ | $(y_b + y_c)/2$ | $(y_b + y_c)/2 + \Delta$ |
|-----------------------|--------|----------|--------|----------------|-----------------|--------------------------|
| accuracy              | 0.8095 | 0.8173   | 0.8773 | 0.8810         | 0.8665          | 0.8775                   |

In this task, the accuracy of all white-box models $y_c$ and $y_{c'}$ are significantly worse than the black-box model $y_b$. Thus the ensembling $(y_b + y_c)/2$ even hurt the performance and $(y_b + y_c)/2 + \Delta$ is no longer be the best. In this case, $y_b + \Delta$ becomes the best one. We suggest applying the transfer editing alone when the black-box model is significantly better than the white-box models, since in this case, ensembling is not working anymore. Last, we report the average accuracy among all 11 tasks in Table 20.

Table 20: The average testing accuracy of 11 tasks when the white-box models are training by 5% training data.

| Average on all tasks | $y_c$ | $y_{c'}$ | $y_b$ | $y_b + \Delta$ | $(y_b + y_c)/2$ | $(y_b + y_c)/2 + \Delta$ |
|----------------------|--------|----------|--------|----------------|-----------------|--------------------------|
| accuracy             | 0.8034 | 0.8383   | 0.8655 | 0.8806         | 0.8704          | 0.8924                   |

In this less training data setting, the improvement by transfer editing which is $y_b + \Delta$ is larger than the performance by ensembling which is $(y_b + y_c)/2$.

## D.8 MIXED DATA EXPERIMENT

In this experiment, 90% keyboard images have a cat as the spurious concept and the other 10% images are unbiased. We include another baseline called **weighted loss**. Specifically, we train a black-box model where the loss of biased and unbiased data has lower (0.1) and higher (0.9) weights, respectively. We report the testing accuracy of methods in Table 21. Results show that transfer editing outperforms weighted loss, demonstrating that transfer editing is more effective in fixing biased models. Note that transfer editing is designed to handle the more challenging setting where no unbiased data are available. We also perform detecting experiments in this setting. The average rank of "cat" is 7 which means our detecting method is also effective in this task.

Table 21: The testing accuracy of methods when using mixed training data.

|  | $y_c$ | $y_c'$ | $y_b$ | transfer edit +ensemble | transfer edit | Weight Loss |
|---|---|---|---|---|---|---|
| Average | 0.9004 | 0.9045 | 0.9174 | 0.9273 | 0.9226 | 0.9183 |
| Random seed 1234 | 0.9005 | 0.9045 | 0.917 | 0.9270 | 0.9235 | 0.9165 |
| Random seed 2345 | 0.9005 | 0.9043 | 0.9160 | 0.9270 | 0.9233 | 0.9163 |
| Random seed 3456 | 0.9005 | 0.9045 | 0.9180 | 0.9278 | 0.9218 | 0.9225 |
| Random seed 4567 | 0.9003 | 0.9045 | 0.9178 | 0.9273 | 0.9215 | 0.9155 |
| Random seed 5678 | 0.9003 | 0.9048 | 0.9180 | 0.9273 | 0.9230 | 0.9208 |

## D.9 EXTRA FIGURES AND TABLES IN FIXING EXPERIMENT

Table 22 and Table 23 show the performance when we apply transfer editing to meta-shift tasks under CLIP-ViT-B/32 and CLIP-ViT-L/14 backbones, respectively. No matter which backbone we use, transfer editing can always help to improve the performance of the black-box model, and transfer editing + ensemble is the best.

In Table 24, we built a black-box model by fine-tuning an ImageNet pre-trained Resnet50 instead of training an MLP based on backbone embeddings. Transfer editing always helps, whether we are using a backbone to obtain a black-box model or fine-tuning a black-box model.

Table 25 and Table 26, performed on image style bias and attribute bias tasks, show that white-box and black-box models can use different types of backbones and our fixing algorithm still works. This is because we fix the model by removing the bias, and it does not depend on the backbone we use.

Table 22: The testing accuracy for meta-shift bias tasks. The backbone we used for both white and black boxes is CLIP-ViT-B/32.

|  | $y_c$ | $y_c'$ | $y_b$ | transfer edit +ensemble | transfer edit | $\frac{y_b+y_c}{2}$ | $\frac{y_b+y_c'}{2}$ | $\frac{y_b+y_c+y_c'}{3}$ |
|---|---|---|---|---|---|---|---|---|
| Average | 0.8252 | 0.8525 | 0.8271 | **0.8661** | 0.8500 | 0.8420 | 0.8612 | 0.8556 |
| Task1 | 0.8591 | 0.8926 | 0.8943 | **0.9111** | 0.9058 | 0.8952 | 0.9076 | 0.8982 |
| Task2 | 0.7638 | 0.8295 | 0.7693 | **0.8443** | 0.8199 | 0.7731 | 0.8302 | 0.8087 |
| Task3 | 0.8051 | 0.8247 | 0.7837 | **0.8296** | 0.8021 | 0.8041 | 0.8183 | 0.8189 |
| Task4 | 0.8026 | 0.8117 | 0.7957 | 0.8201 | 0.7963 | 0.8165 | **0.8220** | 0.8247 |
| Task5 | 0.7897 | 0.8443 | 0.7888 | **0.8723** | 0.8642 | 0.8193 | 0.8665 | 0.8525 |
| Task6 | 0.8738 | 0.8785 | 0.8734 | **0.8970** | 0.8781 | 0.8904 | 0.8933 | 0.8938 |
| Task7 | 0.8691 | 0.8810 | 0.8560 | **0.8797** | 0.8688 | 0.8735 | 0.8788 | 0.8791 |
| Task8 | 0.8381 | 0.8579 | 0.8554 | **0.8745** | 0.8645 | 0.8638 | 0.8728 | 0.8692 |

Table 23: The testing accuracy for meta-shift bias tasks. The backbone we used for both white and black boxes is CLIP-ViT-L/14.

| | $y_c$ | $y_c'$ | $y_b$ | transfer edit +ensemble | transfer edit | $\frac{y_b+y_c}{2}$ | $\frac{y_b+y_c'}{2}$ | $\frac{y_b+y_c+y_c'}{3}$ |
|---|---|---|---|---|---|---|---|---|
| Average | 0.8362 | 0.8681 | 0.8369 | **0.8795** | 0.8599 | 0.8514 | 0.8735 | 0.8676 |
| Task1 | 0.8580 | 0.9026 | 0.9049 | **0.9271** | 0.9249 | 0.8978 | 0.9182 | 0.9060 |
| Task2 | 0.7787 | 0.8402 | 0.7803 | **0.8435** | 0.8190 | 0.7877 | 0.8315 | 0.8185 |
| Task3 | 0.8117 | 0.8416 | 0.7872 | **0.8415** | 0.8079 | 0.8066 | 0.8303 | 0.8305 |
| Task4 | 0.8231 | 0.8289 | 0.8030 | **0.8349** | 0.8059 | 0.8295 | 0.8313 | 0.8338 |
| Task5 | 0.7911 | 0.8702 | 0.8009 | **0.8914** | 0.8662 | 0.8185 | 0.8839 | 0.8655 |
| Task6 | 0.8903 | 0.8982 | 0.8998 | **0.9138** | 0.9023 | 0.9063 | 0.9097 | 0.9087 |
| Task7 | 0.8824 | 0.8926 | 0.8773 | **0.9008** | 0.8873 | 0.8937 | 0.9006 | 0.8992 |
| Task8 | 0.8544 | 0.8703 | 0.8421 | **0.8831** | 0.8654 | 0.8713 | 0.8821 | 0.8784 |

Table 24: The testing accuracy of Celeb-A task when we used fine-tuning black-box models.

| white backbone | black fine-tune | $y_c$ | $y_c'$ | $y_b$ | transfer edit +ensemble | transfer edit | $\frac{y_b+y_c}{2}$ | $\frac{y_b+y_c'}{2}$ | $\frac{y_b+y_c+y_c'}{3}$ |
|---|---|---|---|---|---|---|---|---|---|
| ViT-L/14 | Resnet50 | 0.9792 | 0.9854 | 0.8716 | 0.9709 | 0.9076 | 0.9542 | 0.9657 | 0.9782 |
| ViT-B/32 | Resnet50 | 0.9732 | 0.9819 | 0.8716 | 0.9534 | 0.8910 | 0.9369 | 0.9465 | 0.9659 |

Table 25: The testing accuracy for picture style bias tasks under different backbones.

| white backbone | black backbone | $y_c$ | $y_c'$ | $y_b$ | transfer edit +ensemble | transfer edit | $\frac{y_b+y_c}{2}$ | $\frac{y_b+y_c'}{2}$ | $\frac{y_b+y_c+y_c'}{3}$ |
|---|---|---|---|---|---|---|---|---|---|
| | | | Task 9 [Alarm Clock, Bike(Clip Art), Candles, Fan, Trash Can] | | | | | |
| ViT-L/14 | ViT-L/14 | 0.9376 | 0.9520 | 0.9080 | **0.9688** | 0.9208 | 0.9360 | 0.9456 | 0.9472 |
| ViT-B/32 | ViT-B/32 | 0.8984 | 0.9224 | 0.8248 | **0.9296** | 0.8952 | 0.8880 | 0.9048 | 0.9080 |
| ViT-B/32 | ViT-L/14 | 0.8984 | 0.9224 | 0.9080 | **0.9704** | 0.9344 | 0.9368 | 0.9592 | 0.9448 |
| ViT-L/14 | ViT-B/32 | 0.9376 | 0.9520 | 0.8248 | **0.9544** | 0.8600 | 0.9320 | 0.9504 | 0.9392 |
| ViT-L/14 | Resnet50 | 0.9376 | 0.9520 | 0.3848 | 0.9128 | 0.3984 | 0.8736 | 0.9032 | **0.9400** |
| ViT-B/32 | Resnet50 | 0.8984 | 0.9224 | 0.3848 | 0.8568 | 0.4480 | 0.7088 | 0.7912 | **0.8848** |
| | | | Task 10 [Backpack(Real World), Bottle, Chair, Computer, Hammer] | | | | | |
| ViT-L/14 | ViT-L/14 | 0.9846 | 0.9885 | 0.9500 | **0.9885** | 0.9800 | 0.9808 | 0.9885 | 0.9885 |
| ViT-B/32 | ViT-B/32 | 0.9523 | 0.9692 | 0.8969 | **0.9569** | 0.9338 | 0.9354 | 0.9454 | 0.9462 |
| ViT-B/32 | ViT-L/14 | 0.9523 | 0.9692 | 0.9500 | **0.9800** | 0.9692 | 0.9731 | 0.9769 | 0.9654 |
| ViT-L/14 | ViT-B/32 | 0.9846 | 0.9885 | 0.8969 | **0.9885** | 0.9515 | 0.9731 | 0.9808 | 0.9846 |
| ViT-L/14 | Resnet50 | 0.9846 | 0.9885 | 0.4108 | 0.9769 | 0.4346 | 0.9615 | 0.9692 | **0.9808** |
| ViT-B/32 | Resnet50 | 0.9523 | 0.9692 | 0.4108 | 0.9308 | 0.4469 | 0.9046 | 0.9192 | **0.9377** |

Table 26: The testing accuracy for the attribute bias task under different backbones.

| white backbone | black backbone | $y_c$ | $y_c'$ | $y_b$ | transfer edit +ensemble | transfer edit | $\frac{y_b+y_c}{2}$ | $\frac{y_b+y_c'}{2}$ | $\frac{y_b+y_c+y_c'}{3}$ |
|---|---|---|---|---|---|---|---|---|---|
| ViT-L/14 | ViT-L/14 | 0.9792 | 0.9854 | 0.9852 | **0.9885** | 0.9843 | 0.9868 | 0.9885 | 0.9875 |
| ViT-B/32 | ViT-B/32 | 0.9732 | 0.9819 | 0.9854 | **0.9874** | 0.9866 | 0.9850 | 0.9868 | 0.9849 |
| ViT-B/32 | ViT-L/14 | 0.9732 | 0.9819 | 0.9852 | **0.9883** | 0.9856 | 0.9860 | 0.9876 | 0.9857 |
| ViT-L/14 | ViT-B/32 | 0.9792 | 0.9854 | 0.9854 | **0.9895** | 0.9860 | 0.9876 | 0.9892 | 0.9881 |
| ViT-L/14 | Resnet50 | 0.9792 | 0.9854 | 0.8511 | 0.9758 | 0.8953 | 0.9688 | 0.9756 | **0.9835** |
| ViT-B/32 | Resnet50 | 0.9732 | 0.9819 | 0.8511 | 0.9634 | 0.8817 | 0.9527 | 0.9599 | **0.9745** |

# E  BROADER IMPACTS

While Holmex can detect and remove bias from the model for the benefit of society, there is also the potential for abuse of our editing methods. For example, we could swap the positions of the two

white-box models in the editing section to achieve the goal of injecting more bias into the black-box model. Therefore, our method could be misused to create specific biased models, and we suggest that prevention in this area starts with increased detection of malicious models to avoid biased and unfair models in society.

## F    LIMITATIONS

Our method relies on CLIP, but CLIP is not a reliable representation of some specific concepts, such as some very specialized words in the healthcare domain. To solve this problem, we could perhaps use some vision-language models constructed for specific domains to overcome this limitation.

We also have some limitations in the white box model selection part of the model fixing. For example, a concept may be helpful for a particular class A, but harmful for another class B. If we choose to eliminate the effect of that concept at this point, we will benefit from eliminating the harmful effect on class B, but we will also eliminate the helpful part of that concept for class A.

## G    COMPUTE

All the experiments including calculating the image embeddings are conducted on a *M1 Pro* chip and the running time for every experiment is less than 1 hour.

## H    MORE DISCUSSION ON RELATE WORKS

For detecting spurious correlation, we illustrate the difference between our method and setting from others in Table 27. The first advantage of our method is that we can work for all data containing spurious correlations. We care about this setting since it is a realistic problem and has not been considered by those studies of invariant learning. The second advantage is we can conduct class-level detection instead of instance-level which means we detect a group of images instead of a single one. In addition, our method can provide high-level conceptual interpretation and does not require test data.

| Method | Presentation of spurious correlation | Work for all images contain spurious correlation | conduct class-level detection | concept based method | Do not need test data |
|--------|--------------------------------------|--------------------------------------------------|-------------------------------|----------------------|-----------------------|
| PCBM, (Yuksekgonul et al., 2022) **Ours** | A list of concepts with correlations strength | ✓ | ✓ | ✓ | ✓ |
| Saliency Map based (Singla & Feizi, 2022) (Itti et al., 1998) | A hit map for a given image | ✓ | ✗ | ✗ | ✓ |
| prototypical part network(Chen et al., 2019) | A set of prototype images | ✓ | ✗ | ✗ | ✓ |
| DISC(Wu et al., 2023) | Concept's variance on different environment | ✗ | ✓ | ✓ | ✓ |
| Conceptual Counterfactual Explanation (Abid et al., 2022) | the importance of concepts in a given image | ✓ | ✗ | ✓ | ✗ |

Table 27: Methods for Spurious Correlation Detection

For model fixing, we illustrate the difference between our method and setting from others in Table 28. There are also some advantages of our method. The first one is that our method works for all data containing spurious correlations. The second advantage is that we can fix any black-box model without knowing its weight or retraining it. Last but not least, we do not need any test data.

| Method | Fixing method | Work for all images contain spurious correlation | Can fix any black-box models | Don't retrain black-box models | Do not need test data |
|---|---|---|---|---|---|
| **Ours** | transfer editing | ✓ | ✓ | ✓ | ✓ |
| DISC (Wu et al., 2023) | modify training set | ✗ | ✓ | ✗ | ✓ |
| Instance Reweighting based (Yaghoobzadeh et al., 2021) | reweighing training data | ✗ | ✓ | ✗ | ✓ |
| CBM, PCBM (Koh et al., 2020) (Yuksekgonul et al., 2022) | modify the weights in a linear layer | ✓ | ✗ | - | ✓ |
| Concept-level debugging of part-prototype networks (Bontempelli et al., 2023) | modify the weights in a linear layer | ✓ | ✗ | - | ✓ |
| Last layer re-train (Kirichenko et al., 2023) | re-train the model with extra data set | ✓ | ✓ | ✗ | ✗ |

Table 28: Methods for Model Fixing

