# OpenReview forum: "Holmex: Human-Guided Spurious Correlation Detection and Black-box Model Fixing"
_ICLR.cc/2024/Conference — Submitted to ICLR 2024_

### Official Review · Reviewer_cHZ2 · 2023-11-01

**Soundness:** 3 good
**Presentation:** 2 fair
**Contribution:** 3 good
**Rating:** 6
**Confidence:** 4

**Summary:**

The paper proposes an approach for detected spurious correlations in datasets and transfer editing to transfer the interpretable knowledge to a black box model and improve model accuracy on various tasks. The paper discussed how to detect spurious correlations between concepts in the images and class labels using vision language models such as CLIP. It proposes to train 2 white box models based on frozen CLIP model’s autoencoder backbone and trainable MLP layer the weights of which represent the interpretable importance scores of the concepts (similar to TCAV). One of the white-box models contains the spurious concepts and the other one doesn’t. The differences between 2 white box models are then transferred to the black-box model.
The authors conduct multiple experiments to show the effectiveness of their approach.

**Strengths:**

1) The paper addresses an important problem of identifying and fixing spurious correlations in vision models.
2) It discusses the challenge of entangled concepts and proposes a technique to improve disentanglement using a baseline/neutral concept such as the concept of others.
3) The paper performs through experimentation for different types of spurious correlations (Co-occurrence, Picture style and class attribute).

**Weaknesses:**

1) The paper has multiple important contributions but they are a bit intertwined. In some cases it sounds that the authors use the term bias when they refer to spurious correlations. It would be good to make the terminology consistent and clear.
2) Overall I think that it is a bit hard to follow the paper in terms of understanding the full picture. There are multiple models involved and figure 1 attempts to explain it but it is unclear what bias is and what `compare weights with concept vectors` really means.
3) It is unclear how the human is involved in the guiding of spurious correlation detection and model fixing. It seems that according to the algorithm listing 1, the output of the algorithm is presented to humans but it is unclear how humans guide the process as the title of the paper suggests.
4) In Figure 4 it is unclear how we decide to incorporate the spurious example C_cat into the white-box. How is human involved in that process ?

Minor

eep learning models -> deep learning models

**Questions:**

1) How scalable is the proposed method  ?
2) Is accuracy the main metric used in evaluation experiments ?
3) Why are the experimental results mainly focused on showing the advantage for ensemble models ?

---

> ### Author Response · Authors · 2023-11-22
>
> **Questions in weakness**
>
> The definition of bias in our experiments can be found in the section of experiments, and the `compare weights with concept vectors' part is described in detail in section 4. Algorithm 1 presents a list like Figure 10 in the Appendix to humans, then humans will check if there are spurious correlations. After humans decide which concept is a spurious correlation, the fixing method in Figure 4 can be applied.
>
> **Q1: How scalable is the proposed method**
>
> We applied our method on a large dataset of 40k images (the Celeb-A task), and the running time of the whole method is within 10 mins.
>
> **Q2: Is accuracy the main metric used in evaluation experiments**
>
> Yes, it is.
>
> **Q3: Why are the experimental results mainly focused on showing the advantage of ensemble models?**
>
> People may think that Ensembling is very similar to Transfer Editing as they both do arithmetic among different models and can be helpful for OOD settings. Therefore, we conduct experiments to show the difference between the two methods as well as how we can combine the two methods together to further improve the performance.

---

### Official Review · Reviewer_xRBd · 2023-11-01

**Soundness:** 1 poor
**Presentation:** 1 poor
**Contribution:** 1 poor
**Rating:** 3
**Confidence:** 4

**Summary:**

The paper introduces Holmex for detecting and mitigating spurious correlations based on concept vectors. In spurious correlation detection, the method is based on CLIP and makes two contributions: (1) subtracting a background concept vector (Section 4.1.2) and (2) proposing a new algorithm for stable detection of spurious correlation. In spurious correlation mitigation, the paper proposes transfer editing to mitigate spurious correlations in a black box model. The experiments are conducted on multiple datasets and tasks to show Holmex’s performance on spurious correlation detection and mitigation.

**Strengths:**

* The paper studies an important problem.
* The code is released for better reproducibility.

**Weaknesses:**

## Concerns about the method

### Subtracting background concept vector (Section 4.1.2)
First, I am confused by the motivation of this part of the method. I understand the argument that irrelevant concepts (e.g., cat and airplane) have high cosine similarities. However, I am completely lost for the “model editing experiment” where “a linear layer after the concept activation layer” was trained. Such as the model editing experiment was not introduced before and the details are completely left to Appendix A.1, which was not clear to me either. Second, I wonder why not a simple alternative solution would not suffice. Following the equation on the bottom of Page 4, we can compute
$
P(y = c \mid z) = \frac{\exp(t_c^\top z / T) }{\sum_{c' \in C} \exp(t^\top_{c'} z / T )}
$
, where $T$ is temperature in softmax, $c$ is one concept, and $\mathcal{C}$ is the set of all concepts. You can choose a low temperature to reduce the similarity among different concepts.

### Transfer difference of logits (step 2 in Section 5.1, page 7)
Different models (i.e., white-box and black-box) can have different scales in logits. Although the paper has a discussion of “The scale of logits” on page 7, my question is still not answered.

## Concerns about the experiments

### Datasets and Metrics
I appreciate the authors' efforts in doing experiments for three types of biases. However, I don’t think the paper explains the motivation for creating new evaluation settings and metrics. There are many previous benchmarks and evaluation settings for both (1) spurious correlation detection ([1,2] and (Wu et al., 2023)) and (2) bias mitigation benchmarks ([3-6]).

### Comparison Methods
The proposed method is only compared with a limited number of methods. For spurious correlation detection, the paper is only compared with the PCBM method and its variants. Why not compare with DISC (Wu et al., 2023), which is also a concept-based method? For spurious correlation mitigation, many methods, especially methods that do not rely on concept vectors [1,2,7-9], are not compared.


## References

[1] Sabri Eyuboglu, Maya Varma, Khaled Kamal Saab, Jean-Benoit Delbrouck, Christopher Lee-Messer, Jared Dunnmon, James Zou, and Christopher Re, “Domino: Discovering Systematic Errors with Cross-Modal Embeddings,” in ICLR, 2022.

[2] Gregory Plumb, Nari Johnson, Angel Cabrera, and Ameet Talwalkar, “Towards a More Rigorous Science of Blindspot Discovery in Image Classification Models,” TMLR, 2023.

[3] Nanyang Ye, Kaican Li, Haoyue Bai, Runpeng Yu, Lanqing Hong, Fengwei Zhou, Zhenguo Li, and Jun Zhu, “OoD-Bench: Quantifying and Understanding Two Dimensions of Out-of-Distribution Generalization,” in CVPR, 2022.

[4] Shiori Sagawa*, Pang Wei Koh*, Tatsunori B. Hashimoto, and Percy Liang, “Distributionally Robust Neural Networks for Group Shifts: On the Importance of Regularization for Worst-Case Generalization,” in ICLR, 2020.

[5] Zhiheng Li, Ivan Evtimov, Albert Gordo, Caner Hazirbas, Tal Hassner, Cristian Canton Ferrer, Chenliang Xu, and Mark Ibrahim, “A Whac-A-Mole Dilemma: Shortcuts Come in Multiples Where Mitigating One Amplifies Others,” in CVPR, 2023.

[6] Robik Shrestha, Kushal Kafle, and Christopher Kanan, “An Investigation of Critical Issues in Bias Mitigation Techniques,” in WACV, 2022.

[7] Junhyun Nam, Hyuntak Cha, Sungsoo Ahn, Jaeho Lee, and Jinwoo Shin, “Learning from Failure: Training Debiased Classiﬁer from Biased Classiﬁer,” in NeurIPS, 2020.

[8] Evan Z. Liu, Behzad Haghgoo, Annie S. Chen, Aditi Raghunathan, Pang Wei Koh, Shiori Sagawa, Percy Liang, and Chelsea Finn, “Just Train Twice: Improving Group Robustness without Training Group Information,” in ICML, 2021.

[9] Elliot Creager, Joern-Henrik Jacobsen, and Richard Zemel, “Environment Inference for Invariant Learning,” in ICML, 2021.

**Questions:**

I expect the authors to address my concerns in the response:

1. Why not use software with temperature to address the problem of irrelevant concepts with high similarity?
2. Do you assume that white-box and black-box models share a similar logit scale? If so, this approach is not generalizable enough to claim the “black-box model fixing.”
3. Why create new evaluation settings with new metrics?
4. Add experiments to compare with a broader range of methods.

---

> ### Author Response · Authors · 2023-11-22
>
> **Q1: Why not use softmax with temperature to address the problem of irrelevant concepts with high similarity?**
>
> We didn't understand what you meant by using temperature to address the issue. The irrelevant concepts with high cosine similarity will be an issue when we represent the concept with this raw text embedding and capture the concept strength by inner-product with raw text embedding with image embeddings. It is not an issue in the original CLIP classifier. We calculate the similarity by cosine-similarity, its range is from -1 to 1. If using cosine-similarity divided by $T$, then its range will change from $-1/T$ to $1/T$. The similarity does not depend on the absolute magnitude. It depends on the relative magnitude.
>
> **Q2: Do you assume that white-box and black-box models share a similar logit scale?**
>
> There is no free lunch for the black-box model fixing.
> Consider a black-box model that just randomly outputs results. No method could cure this black-box model, and the only thing we can do is ignoring its outputs. One important intuition is that a normal model should output a proper probability distribution over class labels. For example, if there are 80\% rainy samples and 20\% sunny samples for the same given input $x$ in the training set, then a normal model will output a probability distribution similar to $(0.8, 0.2)$. Since the mapping between probabilities and logits is bijective (if we ignore the constant shift), we can derive that the logits are roughly $(1.375+constant, 0+constant)$ based on the probabilities $(0.8, 0.2)$. Thus, a normal model's logits can be roughly decided by the proper probabilities. As a result, if the model does not output probabilities too different from the proper ones, applying $y_b + \Delta$ should keep logits on the right scale.
>
> **Questions about Comparison Methods**
>
> We discuss the difference between DISC and our method, why we cannot compare it under our setting, and why we raise our setting. Those discussions are in the comment reply of comparing to DISC. We add a new section in Appendix H to illustrate the difference between our method and the others. We highly recommend reading those tables in Appendix H for a better understanding. As for those non-concept-based methods, we cannot make a fair and reasonable setting, neither. We use the example of salience-map-based methods to illustrate the difficulty and please refer to our reply to review NL8r for this part.

---

> > ### Comment · Reviewer_xRBd · 2023-11-23
> >
> > ### Subtracting background concept vector
> > Note that temperature is used in CLIP pertaining (see "scaled pairwise cosine similarities" in Figure 3 "Numpy-like pseudocode for the core of an implementation of CLIP." in the original CLIp paper [10]). Thus, it is reasonable to adjust the scale of the cosine similarity so that the similarity of unpaired concepts can be reduced. Although the authors argue that the scale of the cosine similarity will change after applying temperature, it is noteworthy that the ultimate goal is to compute the score between the concept of an image embedding, where the scores can still be more discriminative among different concepts after applying the temperature.
> >
> > ### Scale of logits
> > I appreciate the authors' explanation of the scale of logits, which addressed my concerns.
> >
> > ### Benchmark and Comparison Methods
> > I partially agree with the authors on the difficulties in comparing against previous concept-based bias detection methods on bias detection benchmarks. However, I still believe that it is necessary to compare with previous bias mitigation methods, including ones without relying on concepts [1,2,7-9] on standard bias mitigation benchmarks [3-6].
> >
> > Since I still have unaddressed concerns, I keep my rating as "3: reject."
> >
> > #### References
> >
> > [10] Learning Transferable Visual Models From Natural Language Supervision. https://arxiv.org/abs/2103.00020

---

### Official Review · Reviewer_NL8r · 2023-11-01

**Soundness:** 2 fair
**Presentation:** 2 fair
**Contribution:** 1 poor
**Rating:** 3
**Confidence:** 4

**Summary:**

The paper introduces Holmex, a method designed for human-guided spurious correlation detection and black-box model fixing. It enables humans in the deep model debugging process by addressing two main tasks:

Detecting Spurious Correlations: Holmex uses pre-trained vision-language models to create separable vectors representing high-level and meaningful concepts. It proposes a novel algorithm based on these concept vectors to detect conceptual spurious correlations in training data, and this algorithm is more stable than previous methods.

Fixing Biased Black-Box Models: Unlike prior approaches that focus on making biased models interpretable and editable, Holmex is compatible with arbitrary black-box models. It introduces a novel technique called "transfer editing" to transfer revisions made in interpretable models to correct spurious correlations in black-box models.

**Strengths:**

The strengths of the paper are as follows:

1. Improved Concept Embeddings: The paper enhances the quality of concept embeddings by reducing the entanglement of raw text embeddings. It achieves this by subtracting a vector of the background word, which is a useful contribution. This improvement is crucial for accurate detection of spurious correlations.

2. Novel Detecting Algorithm: The paper introduces a novel detecting algorithm that is specifically designed to reveal correlations between concepts and labels in a stable manner. This algorithm enhances the reliability and stability of the spurious correlation detection process.

3. Transfer Editing Technique: The paper proposes a transfer editing technique, which is a novel method for transferring revisions made by humans in white-box models to black-box models. This approach enables the fixing of spurious correlations in black-box models, making it a versatile and impactful contribution.

The paper conducts extensive experiments on multiple datasets with different types of biases, including co-occurrence bias, picture style bias, and class attribute bias. This demonstrates the effectiveness and applicability of the Holmex method across a range of real-world scenarios, which is a significant strength in showcasing its practical utility.

**Weaknesses:**

The paper does not cite several works in this domain. Some of the missing citations are:

1. Salient ImageNet: How to detect spurious correlations in deep learning? ICLR 2022.
2. Last Layer Re-Training is Sufficient for Robustness to Spurious Correlations. ICLR 2023.
3. Wilds: A benchmark of in-the-wild distribution shifts. PMLR, 2021.

Salient ImageNet provides a scalable methodology for identifying spurious correlations at scale. The paper does not include any comparison with that method, rather no citation is provided. Similarly, the latter paper provides a method for robustifying against spurious correlations. Again, no citation provided.

This provides a strong evidence that the paper is written without a thorough research of the prior work.

**Questions:**

There is no comparison against several of the group robustness methods presented in the prior works. Given the extent of the literature on robustness against spurious correlations, results comparing the accuracy of the proposed method against the baseline are not acceptable.

---

> ### Author Response · Authors · 2023-11-22
>
> We have cited all the mentioned papers in the revision paper, and we added a new section in Appendix H to illustrate the difference between our method and the others. We believe reading those tables in Appendix H can help better understand the difference. For the task setting in Wilds, we discuss the difference in the common reply about DISC. Below, we use the salience-map-based method as an example to show its limitations and why we cannot conduct the comparison.
> We have done thorough research on the prior work and those discussions about the salience map can be also found in related work.
>
> **Limitation of salience map and why we cannot compare with it**
>
> \noindent
> The first limitation of the salience map is that it cannot bring high-level interpretations. Take the fire truck in Canberra as an example. Since the fire trucks in Canberra are in yellow, `yellow' becomes a spurious correlation when we apply the model to other cities. If we use a salience map for interpretation, it might focus on the fire truck's body. However, we cannot tell the detailed reason why the model pays attention to it. The reason might be the color, texture, or other things that make the model see the fire truck's body. Thus, it is hard to evaluate if it is correct when a salience map shows the truck body. Another limitation is that the salience map only interprets a single image instead of a class. In contrast, our method can provide class-level interpretations. Those differences make the comparison between our method and the salience map not appropriate in a fair and reasonable setting.

---

> > ### Comment · Reviewer_NL8r · 2023-12-04
> > **Not satisfied with the author response**
> >
> > I disagree with the assertion that the method of interpretation introduced in the Salient ImageNet paper cannot be used to identify color-like spurious correlations. The visualization methodology in the Salient ImageNet (which was originally introduced in the paper: https://arxiv.org/abs/2012.01750) also includes feature attack. Here, you can change the original image so that the changed image can be used to identify such correlations.
> >
> > Moreover, the salient imagenet methodology is used to perform analysis at the class level (not at a single image level). That is precisely why a comparison with the paper should have been included.

---

### Official Review · Reviewer_G5F3 · 2023-11-06

**Soundness:** 3 good
**Presentation:** 3 good
**Contribution:** 3 good
**Rating:** 6
**Confidence:** 4

**Summary:**

This paper presents HOLMEX, a method to identify a model's reliance on spurious correlations, and fix it. The general idea is to construct concept vectors using a large pre-trained model like CLIP, and then surface the correlation between a concept vector and a label to a human. The human can then use their inductive bias/domain knowledge to trim correlations that happen to be spurious. Once the spurious concept and label tuple is detected, the authors then propose a transfer technique to edit the model. In the transfer editing technique, you train two 'white-box' models: 1) where the spurious concept has been removed, and 2) where the spurious concept is present. These whitebox concepts are basically softmax linear layers on top on the clip representations for the input samples. The hope here is that the weights of these two whitebox models capture the spurious direction. To perform transfer editing, you take a difference between the logits of the two whitebox models, and add that to the logits of the blackbox model. They couple this approach with ensembling and show that such an approach leads to improved model performance.

**Strengths:**

Overall, this papers sets out an important problem and presents a scheme for addressing that problem. I list below some nice aspects of this work:

- **Modular Approach**: The paper separates detecting spurious correlation from fixing it.
- **Clear scheme**: The paper describes its scheme very clearly, and tries to justify each step of the scheme.
- **Demonstrates performance improvement**: The paper also shows that the transfer editing schemes and ensembling leads to improved performance across the board across all the tasks tested.
- **Control experiment**: I liked that the authors included a control experiment for a model with no spurious signals. The approach shows the kind of null behavior you would want in that setting.

**Weaknesses:**

Below I discuss some of the weaknesses of the scheme presented here.

- **Too many moving parts**: While the scheme presented is modular. As it stands, there are several decisions that need to be gotten right for the overall scheme to work. Here is what I mean: 1) it looks like the traditional concept vectors (derived from model embeddings) are ineffective, so we need a modified version, 2) One needs a background word, 3) One needs to train a linear classifier to estimate correlations, 4) One needs to train two separate linear classifiers again to do editing for each spurious concept that you want to remove. This means that if you have 20 concepts to remove, then you would be training 40 linear classifiers to remove the effect of these 20 spurious concepts for that label alone. If any of the steps that I have listed does not work, then the entire scheme does not work.

- **Over reliance on CLIP**: I think the dependence on CLIP in this work is quite worrisome. I think the CLIP embeddings are effective probably because the CLIP dataset is quite large, so those embeddings don't suffer from the issues the authors noticed. For example, imagine that you wanted to now fix a model that solely relies on CLIP embeddings as its classifier, then I assume the approach here would be ineffective?

- **Logit Correction in Transfer Editing**: I am surprised that the editing scheme here works since we can simply think of this as shifting the distribution of the logits. However, it requires that the output space for the black-box models be the same size as that of the model you want to edit.

**Questions:**

Here are some questions for the authors:
- How do you think the transfer editing approach here relates to the task vectors approach? See: Editing models with Task Arithmetic, and Task Arithmetic in the Tangent Space. It seems like you could avoid training two white-box models by adopting the task arithmetic editing approaches in the above papers.

- What is the justification for why the concept vectors from raw embeddings does not work? What if I have a model that just uses the clip embedding itself for classification, but the clip embedding has spurious correlations too? Is this approach just inheriting the limitations of clip representations? What about if I have satellite images or a setting where CLIP is not useful?

- Ensembling: Did you test ensembling alone in Table 3? It would be interesting to see whether simply ensembling recovers the performance gains that you see in that table. I ask this because ensembling has been shown to give OOD benefits.

---

> ### Author Response · Authors · 2023-11-22
>
> **Question in weakness: If we have 20 concepts to remove**
>
> For this case, you can still train 2 white-box models, one contains those 20 concepts and one does not. Thus, you don't need to train 40 models.
>
> **Q1: Transfer editing vs Task Arithmetic**
>
> One similarity between transfer editing and task arithmetic is that they both do some meaningful arithmetic combination. One major difference between transfer editing and task arithmetic is that transfer editing does arithmetic among different models while task arithmetic does arithmetic among different weights of the same model. We think task arithmetic also has a potential in model fixing, but it may have two drawbacks. First, we need extra datasets with annotated concepts. For example, to cure the model 'cat with keyboard', we need two datasets 'dog with keyboard' and 'dog without keyboard'. Second, we need access to the weights of black-box models. Those two requirements are drawbacks of task arithmetic since sometimes fine-tuning or re-training a black-box model can be costly, and accessing the model weights is prohibited for some API-based black-box models. However, transfer editing does not need any extra datasets, and as long as we can access the final output probabilities of black-box models, we can recover their logits without knowing their weights. Thus, from this point of view, transfer editing is very different from task arithmetic.
>
> **Q2: Reliance on CLIP**
>
> Comparing the equation at the end of page 4, the raw embedding replaces $t_{other}$ by a zero vector. We believe the zero vector is not as appropriate as the background vector since CLIP is not trained in that manner. In the case of satellite images where CLIP is not useful, we suggest using some related datasets or vision-language contrastive models to obtain meaningful concept vectors. So when CLIP does not work well, we just resort to other methods to obtain an off-the-shelf model. We want to emphasize again that in our paper we focus on how to extract concepts from off-the-shelf models.
>
> **Q3: More ensembling results**
>
> Yes, you can find those results in Table 22 and Table 23 in Appendix.

---

> > ### Comment · Reviewer_G5F3 · 2023-11-22
> > **Thanks**
> >
> > Thanks to the authors for their response. I am inclined to keep my current rating. Overall, my chief concern about the complexity of the proposal remains. I'll take some time to digest the rest of the feedback.

---

### Author Response · Authors · 2023-11-22
**Common Reply: brief introduction to DISC and its limitation**

**The Assumption on Training Distribution of DISC**

 (quote from the original paper of DISC) ''To guarantee the distinguishability of spurious concepts, we assume the training distribution is representative of the overall distribution. For example, if ''bed'' always coexists with ''cat'' in the training dataset, then there is no way we can distinguish that ``cat'' is the invariant concept while ''bed'' is not.''

**How DISC detects spurious correlation**

DISC maps the images of one class into an embedding space of a neural network that is trained by Empirical Risk Minimization. Then it clusters those embeddings into $k$ groups. The Concept Tendency Score (CTS) of a concept is defined as the variance of the concept among those $k$ clustering groups. For example, DISC assumes those $k$ clustering groups as $k$ different environments of 'cat'. Those $k$ environments might be cats in the bedroom, cats in the kitchen, cats outdoors, and so on. If we consider the concept 'cat', since all environments contain cats, the CTS, or variance, will be small. Considering the concept 'bed', the CTS of 'bed' will be larger. Thus, DISC can detect the spurious correlation by checking the concepts with large CTS.

**DISC's limitation and why it can't solve our setting**

DISC fully relies on the assumption that the spurious correlation varies in different environments in the training data. While in our setting where all images of 'cat' contain 'keyboard', the spurious correlation 'keyboard' does not vary in the training set. Thus, DISC will also bring a small CTS and cannot detect the spurious correlation 'keyboard'.

**Why the setting of all images in a class contains spurious correlation important**

First, considering the setting that all data samples contain bias will broaden the setting of Invariant Learning. It is a new challenge that is out of the assumption of DISC.
Second, this is indeed a realistic setting. In fact, we cannot always assume the availability of an extensive perfect dataset that contains all situations. Let's consider an example in which we collect a dataset consisting of vehicle photos in Canberra. When we train a model on it and apply it to other cities, it might contain a spurious correlation for the class 'fire truck'. It is because all fire trucks in Canberra are in yellow, and fire trucks are in red in other cities. In this case, all fire truck photos contain spurious correlations. As another example, if we collect data from a hospital in China, then all the patients are Chinese and there might be a spurious correlation about race.

---

### Meta-Review · Area_Chair_PFkY · 2023-12-06

**Metareview:**

The paper studies the two-part problem of (1) detecting and (2) removing the spurious features in image classification. The first part of the method identifies concepts correlated with the class label. This is done using a frozen CLIP encoder for both text and images. The authors provide a method for removing a baseline "background concept vector" to enhance this concept identification. Then, a human in the loop identifies which of the concepts are spurious. Finally, two white-box models are trained: one relying on the spurious concept and one ignoring it. The authors mix these models with a black-box model trained directly on the images, and use it as a corrected predictor.

## Strengths

The paper addresses an important problem. Detecting spurious features is extremely challenging, and human-in-the-loop approaches are needed.

The proposed method is well-motivated, and the experiments show promising results.

## Weaknesses

Multiple reviewers flagged that the paper is hard to follow, and that they were confused about the details of the method. It was also highlighted that the method has many moving parts and may be hard to get right in practice.

Most importantly, the paper uses a non-standard evaluation. The authors use non-standard datasets and are missing important baselines from prior work.

**Justification For Why Not Higher Score:**

The reviewers remained unsatisfied with the evaluation of the proposed method and the comparison to baselines. It is challenging to interpret the results and put them in the context of the current literature.

**Justification For Why Not Lower Score:**

N/A

---

### Decision · Program_Chairs · 2024-01-16

Reject